# Torsin ATPases influence chromatin interaction of the Torsin regulator LAP1

**Naemi Luithle[1], Jelmi uit de Bos[1,2], Ruud Hovius[3], Daria Maslennikova[1,2], Renard TM Lewis[1,2], Rosemarie Ungricht[1], Beat Fierz[3], Ulrike Kutay[1]***

[1]Institute of Biochemistry, Department of Biology, ETH Zurich, Zurich, Switzerland; [2]Molecular Life Sciences Ph.D. Program, Zurich, Switzerland; [3]Institute of Chemical Sciences and Engineering - ISIC, EPFL, Lausanne, Switzerland

**Abstract** The inner nuclear membrane is functionalized by diverse transmembrane proteins that associate with nuclear lamins and/or chromatin. When cells enter mitosis, membrane-chromatin contacts must be broken to allow for proper chromosome segregation; yet how this occurs remains ill-understood. Unexpectedly, we observed that an imbalance in the levels of the lamina-associated polypeptide 1 (LAP1), an activator of ER-resident Torsin AAA+-ATPases, causes a failure in membrane removal from mitotic chromatin, accompanied by chromosome segregation errors and changes in post-mitotic nuclear morphology. These defects are dependent on a hitherto unknown chromatin-binding region of LAP1 that we have delineated. LAP1-induced NE abnormalities are efficiently suppressed by expression of wild-type but not ATPase-deficient Torsins. Furthermore, a dominant-negative Torsin induces chromosome segregation defects in a LAP1-dependent manner. These results indicate that association of LAP1 with chromatin in the nucleus can be modulated by Torsins in the perinuclear space, shedding new light on the LAP1-Torsin interplay.

***For correspondence:**
ulrike.kutay@bc.biol.ethz.ch

**Competing interests:** The authors declare that no competing interests exist.

## Introduction

In all eukaryotes, a double membrane barrier termed nuclear envelope (NE) serves as the boundary of the nuclear compartment that safeguards the genetic information. The NE is built by a large chromatin-attached membrane sheet of the endoplasmic reticulum (ER). Despite the connectivity of the NE-ER membrane network, the inner nuclear membrane (INM) contains a unique set of transmembrane proteins (*Holmer and Worman, 2001*; *Ungricht and Kutay, 2015b*). In general, enrichment of these membrane proteins from the peripheral ER at the nuclear face of the NE relies on several domains or short linear motifs in their extralumenal domains that together ensure retention on nuclear partners such as nuclear lamins, chromatin-associated factors or DNA (*Boni et al., 2015*; *Powell and Burke, 1990*; *Ungricht et al., 2015a*). While the interaction of INM proteins with the nuclear lamina is restricted to lamin-expressing metazoan cells and certain protists, chromatin is the principal binding partner of INM proteins in all eukaryotes. The association of INM proteins with chromatin is functionally important, being it for the formation of gametes, for development or differentiation, exemplified by the NE-based pairing of homologous chromosomes during meiosis, or the progressive enrichment of transcriptionally repressed chromatin domains at the nuclear periphery of differentiating metazoan cells (*Ungricht and Kutay, 2017*).

How INM proteins associate with chromatin is best understood for cases in which folded domains are used for chromatin binding. The lamin B receptor (LBR), for example, contains an N-terminal tudor domain for interaction with the epigenetically marked tail of histone H4 as well as structurally less-defined regions for association with DNA and HP1 (*Hirano et al., 2012*). Some INM proteins contain short bihelical motifs for chromatin interaction such as the LEM (LAP2-emerin-MAN1) domain, which binds the chromatin factor BAF, or related motifs that bind DNA directly (*Barton et al., 2015*; *Brachner and Foisner, 2011*). The interaction of other mammalian INM

proteins with chromatin is less understood. Biochemical experiments suggest that many INM proteins might even interact with DNA directly (*Ulbert et al., 2006*).

In cells undergoing open mitosis, the entire nuclear compartment is disintegrated by NE breakdown (NEBD), which liberates chromosomes from confinement by the nuclear membrane (*Champion et al., 2017*). In the course of prophase, the multifaceted interactions between INM proteins, chromatin, and nuclear lamins are broken, allowing for the separation of membranes from chromatin and the partitioning of INM proteins into the mitotic ER network (*Ellenberg et al., 1997*; *Yang et al., 1997*). Exploiting a synthetic membrane–chromatin tethering system, we recently demonstrated that failed removal of membranes from chromatin impairs mitotic chromatin organization, chromosome segregation and cytokinesis, and perturbs post-mitotic nuclear morphology (*Champion et al., 2019*). Notably, also in yeasts with closed mitosis, the detachment of chromosomes from the NE is required for faithful chromosome segregation (*Titos et al., 2014*). It is generally assumed that changes in posttranslational modifications of INM proteins and chromatin-associated factors, especially protein phosphorylation, induce membrane dissociation from chromatin during mitotic entry. In strong support of this view, many mammalian INM proteins are phosphorylated in their nucleoplasmic domains by CDK1 or other abundant mitotic kinases during mitotic entry (*Güttinger et al., 2009*). Also chromatin-associated interaction partners of INM proteins such as BAF are released from chromatin by mitotic phosphorylation (*Samwer et al., 2017*). After prophase, multiple mechanisms ensure that the ER does not re-associate with chromatin prematurely, before NE reformation starts in late anaphase (*Champion et al., 2019*).

When studying the release of membrane proteins from chromatin during mitotic entry in cultured human somatic cells, we observed that an increased cellular level of one specific INM protein, the lamina-associated polypeptide 1 (LAP1), caused a failure in membrane removal from chromatin in mitosis. This observation not only indicated that LAP1 is a novel chromatin-binding INM protein but also suggested that some cellular factor(s) might have become limiting for breaking mitotic LAP1-chromatin contacts. LAP1 is an INM-localized activator of Torsins; the only known AAA+ ATPases residing in the lumen of the ER and the continuous perinuclear space (*Brown et al., 2014*; *Goodchild and Dauer, 2005*; *Laudermilch and Schlieker, 2016*; *Sosa et al., 2014*). Although neither the precise biological role nor the substrates of Torsins are known, loss of Torsin functionality causes changes in NE morphology (*Goodchild et al., 2005*), pointing toward a role at the NE. Mutations in *TOR1A*, the predominant Torsin isoform expressed in the brain (*Jungwirth et al., 2010*), lead to a severe movement disorder termed early-onset dystonia (DYT1) (*Ozelius et al., 1997*). Notably, also mutations in the *LAP1* gene, which is ubiquitously expressed in the human body, have been linked to primary dystonia as well as to cardiomyopathy, cerebellar atrophy, and cancer (*Dorboz et al., 2014*; *Fichtman et al., 2019*; *Kayman-Kurekci et al., 2014*; *Rebelo et al., 2015*).

In our experiments, persistence of LAP1 on mitotic chromatin was accompanied by changes in post-mitotic NE morphology and cell division errors, reminiscent of defects that we had previously reported for a synthetic membrane–chromatin tether, which prevents mitotic chromatin release. Strikingly, overexpression of wild-type but not of ATPase-deficient Torsins efficiently suppressed LAP1-induced NE abnormalities. Furthermore, a dominant-negative Torsin induced chromosome segregation defects in a LAP1-dependent manner, suggesting that LAP1 association with chromatin is modulated by Torsin ATPases residing in the perinuclear space and ER lumen. Collectively, our results underscore the importance of dissolving INM protein-chromatin interactions for mitotic fidelity and suggest that LAP1 may not merely be important for Torsin ATPase activation but itself subject to Torsin regulation across the INM.

## Results

### Increased levels of LAP1 impair the dissolution of LAP1–chromatin contacts during mitosis

Although it is generally assumed that changes in posttranslational modifications of INM proteins and chromatin-associated factors, especially protein phosphorylation, induce membrane dissociation from chromatin during mitotic entry, direct evidence in support of this hypothesis in living cells is lacking and our molecular understanding of the process is limited. Here, we set out to gain new insights into the mechanisms required for dissociation of INM proteins from chromatin. In a first

step, we tested whether it is possible to impair the release of the NE membranes from mitotic chromatin if any cellular factor involved in dissolving INM protein-chromatin contacts would become limiting by a disturbed ratio between INM proteins and the potential release factor(s). Thus, we overexpressed several abundant INM proteins, including emerin, SUN1, SUN2, LAP2β, LEM2, and LAP1 in HeLa cells by transiently transfecting the respective expression vectors. Remarkably, overexpression of LAP1B, the longest human isoform of LAP1, led to severe NE aberrations, while overexpression of other INM proteins did not cause similar defects (*Figure 1A*). Interestingly, the LAP1-induced NE aberrations were observed in the vast majority of cells when analyzed after 48 hr, whereas most cells still displayed a normal nuclear morphology after 24 hr, perhaps because more cells had progressed through mitosis in presence of LAP1 at the later time point. Changes in NE morphology upon LAP1B expression were also evident in other cell types such as HCT116 or HepG2 cells (*Figure 1—figure supplement 1A*). The LAP1B-induced NE aberrations were reminiscent of those that we had previously observed using a synthetic membrane–chromatin tethering system that prevents the release of the NE/ER network from chromatin during mitosis (*Champion et al., 2019*).

To analyze whether these NE aberrations indeed originated from failed release of LAP1B and thereby membranes from chromatin during mitosis, we performed time-lapse imaging of cells expressing similar levels of either LAP1B-GFP or LAP2β-GFP (*Figure 1B*, *Figure 1—figure supplement 1B*). Before mitotic entry, nuclei of LAP1B-GFP expressing cells exhibited a normal round shape (*Figure 1B*, *Video 1*). Strikingly, throughout mitosis, LAP1B-GFP remained associated with chromatin, and after NE reassembly, both daughter cells exhibited severe nuclear deformations. In contrast, LAP2β-GFP was faithfully released into the mitotic ER, and post-mitotic nuclei of LAP2β-GFP-expressing cells possessed a round unperturbed morphology (*Figure 1B*, *Video 2*).

To address how the observed phenotype relates to increasing expression levels of LAP1, we used tetracycline-inducible HeLa cells to tune the amount of LAP1B-GFP. In these experiments, we also included LAP1C, the second isoform of human LAP1 (*Figure 1C*). LAP1C is produced from a shorter transcript and starts at an alternative translational start site (Met 122), leading to an isoform that is N-terminally truncated by 121 amino residues (*Figure 1C*; *Santos et al., 2014*). Cells were analyzed by confocal microscopy 48 hr after tetracycline induction (*Figure 1D*), and expression levels were examined by quantification of GFP intensity (*Figure 1E*) and immunoblotting (*Figure 1F*). The severity of the NE aberrations indeed increased with rising expression levels for both LAP1B-GFP and LAP1C-GFP (*Figure 1D, E, F*); however, they were more pronounced for LAP1B when comparing cells with a similar expression level of both isoforms. We also inspected fixed cells for an association of membranes with chromatin during metaphase (*Figure 1G*). Both GFP-tagged LAP1 isoforms were enriched at mitotic chromatin; however, the enrichment of LAP1B-GFP was more prominent, indicating a stronger retention of the longer isoform on mitotic chromatin.

To exclude that persistent LAP1–chromatin contacts affect the mitotic localization of endogenous NE proteins, we fixed uninduced and induced LAP1B-GFP cells, and immunostained for nuclear lamins and multiple INM proteins (*Figure 1—figure supplement 1C*). Analysis of metaphase cells revealed that of the tested INM proteins only LAP1B-GFP was strongly enriched at mitotic chromatin. Taken together, overexpression of both human LAP1 isoforms results in their failed removal from mitotic chromatin and cells display an aberrant post-mitotic nuclear morphology, whereas overexpression of other INM proteins at comparable expression levels does not cause similar phenotypes.

## The molecular architecture of the nucleoplasmic domain of LAP1

LAP1 has originally been identified as an INM protein that interacts with nuclear lamins (*Foisner and Gerace, 1993*; *Senior and Gerace, 1988*), yet the molecular architecture of its nucleoplasmic domain remains poorly characterized. Fluorescence recovery after photobleaching (FRAP) experiments previously revealed an extremely low diffusional mobility of LAP1B at the INM (*Zuleger et al., 2011*), but how it is so strongly immobilized at the INM remained to be determined. As overexpressed LAP1B-GFP was associated with chromatin during mitosis, we reasoned that interaction with both chromatin and nuclear lamins may anchor LAP1B at the INM. Our FRAP experiments confirmed that LAP1B-GFP is a highly immobile INM protein (*Figure 1—figure supplement 2A*). In contrast, LAP1C-GFP was more mobile, showing a recovery curve similar to LAP2β-GFP.

To elucidate which lamins contribute to the retention of LAP1 at the NE, we examined the diffusional mobility of LAP1B-GFP and LAP1C-GFP after RNAi-mediated depletion of lamin A/C, lamin B1, or lamin B2 by FRAP (*Figure 1—figure supplement 2B, C*). Downregulation of lamin A/C and

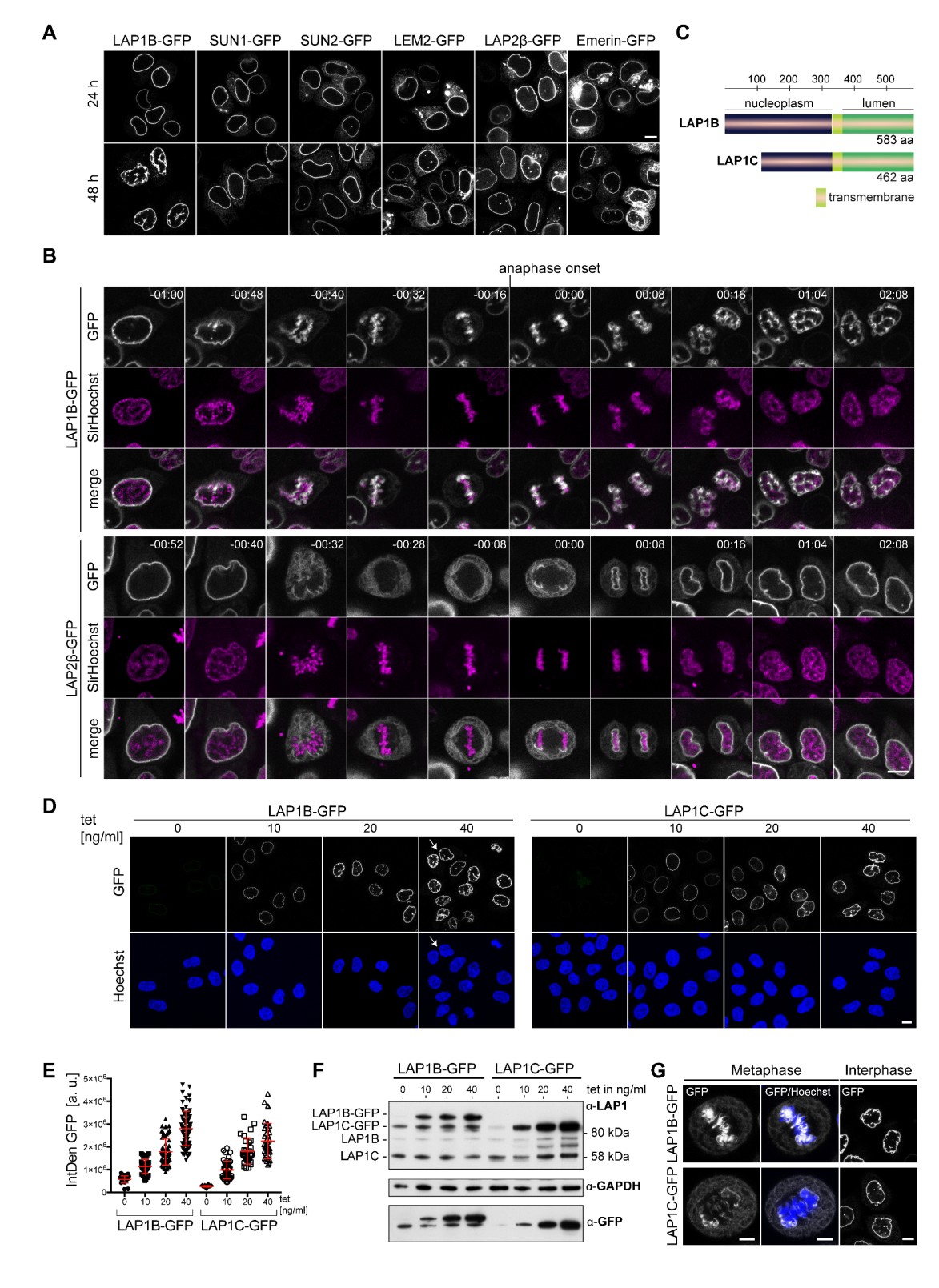

**Figure 1.** Overexpression of LAP1B and LAP1C causes post-mitotic nuclear envelope (NE) aberrations. (**A**) Vectors encoding the indicated inner nuclear membrane (INM) proteins were transfected into HeLa cells. Cells were fixed after 24 or 48 hr and analyzed by confocal microscopy. Scale bar, 10 μm. (**B**) Time-lapse images of LAP1B-GFP or LAP2β-GFP expressing HeLa cells progressing through mitosis. Expression of the constructs was induced 24 hr prior to imaging. DNA was visualized by SirHoechst and used to define anaphase onset (t = 00:00 hr:min). Scale bar, 5 μm. (**C**) Scheme depicting the

*Figure 1 continued on next page*

*Figure 1 continued*

two human LAP1 isoforms, LAP1B and LAP1C. (**D**) Representative images of stable HeLa cells lines expressing LAP1B-GFP and LAP1C-GFP after induction with different tetracycline (tet) concentrations for 48 hr. Please note that we observed some 'twin' nuclei upon overexpression of LAP1B (white arrow), which can be a sign of binucleation, as further analyzed in *Figure 5*. Scale bar, 10 µm. (**E**) LAP1 levels at the NE of cells from the experiment shown in panel D were analyzed by quantification of the integrated GFP density (IntDen GFP) per nucleus. (**F**) LAP1 levels in cell lysates derived from the experiment shown in panel D were analyzed by immunoblotting. Note that the LAP1B-GFP cell line expressed LAP1C-GFP independent of tetracycline addition, due to an alternative transcriptional start site used for the production of the shorter LAP1 isoform (*Santos et al., 2014*). (**G**) Left: Maximum intensity z-projections (5 × 0.63 µm) of confocal images from fixed metaphase HeLa cells expressing LAP1B-GFP or LAP1C-GFP after 48 hr of induction. Scale bar, 5 µm. Right: Confocal images of fixed HeLa cells in interphase. Scale bar, 10 µm.

The online version of this article includes the following figure supplement(s) for figure 1:

**Figure supplement 1.** Interphase and mitotic localization of LAP1B-GFP and endogenous nuclear envelope (NE) proteins.

**Figure supplement 2.** Lamin A/C and lamin B1 contribute to the retention of LAP1B at the nuclear envelope (NE).

---

lamin B1, but not of lamin B2, led to an increase in the diffusional mobility of LAP1B at the NE. The combined depletion of lamin A/C and lamin B1 had an even stronger effect, indicating that both A- and B-type lamins contribute to retention of LAP1B at the INM. For LAP1C, only the loss of lamin A/C had a prominent effect on its mobility, suggesting that A-type lamins play a major role in retention of the shorter isoform at the INM.

The extraluminal domain of LAP1B is predicted to be largely intrinsically disordered (*Figure 2—figure supplement 1A*), and we set out to delineate its chromatin and lamina-binding regions. By truncation analysis, we loosely delineated two regions in LAP1B that are sufficient for lamina association; an N-terminal fragment of 72 amino acids and a second region comprising residues 184–337 (*Figure 2—figure supplement 1B, C, D, E, F*). To define which part of LAP1 interacts with chromatin, we analyzed the mitotic localization of soluble LAP1B fragments fused to GST-GFP (*Figure 2A, B*, *Figure 2—figure supplement 1A, B*). Whereas the N-terminal part (aa 1–183) of LAP1B localized to the metaphase plate, LAP1B(184-337)-GST-GFP was distributed throughout the mitotic cytoplasm. By testing further truncations, we mapped the chromatin-binding region (CBR) of LAP1B to residues 98–136 (*Figure 2B*). This protein fragment was very efficiently enriched on chromatin throughout mitosis (*Figure 2C*) and is sufficient for chromatin interaction. Importantly, deletion of the CBR from full-length, membrane-bound LAP1B-GFP abolished its association with chromatin during mitosis (*Figure 2D*).

Next, we examined whether the CBR contributes to NE localization of LAP1B during interphase. To do so, we chose a domain transfer approach, fusing the CBR of LAP1B to the ER-localized protein SPAG4, as we did before when delineating NE targeting motifs of other INM proteins (*Turgay et al., 2010*). Consistent with our assumption that chromatin-binding contributes to the retention of LAP1B at the INM, the CBR was sufficient to target the CBR-SPAG4-GFP fusion protein to the NE (*Figure 2E*). Finally, we tested whether the CBR can interact with chromatin directly by performing electrophoretic mobility shifts assays (EMSAs), revealing that the purified recombinant CBR can interact with reconstituted nucleosomes and even naked DNA (*Figure 2—figure supplement 2C, D*).

Collectively, our mapping experiments delineated three regions within the disordered nucleoplasmic domain of LAP1B that confer interaction with nuclear components: two regions that confer association with the nuclear lamina (aa 1–72 and 184–337) and a central region that is sufficient for chromatin interaction (CBR, aa 98–138; *Figure 2G*). The multiple regions of LAP1 used for interaction with nuclear partners can very well explain the strong retention of LAP1B at the INM by binding avidity.

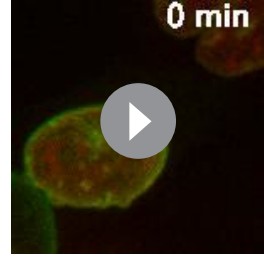

**Video 1.** Time-lapse imaging of a LAP1B-GFP expressing cell progressing through mitosis. Expression of LAP1B-GFP was induced 24 hr prior to imaging. DNA was visualized by SirHoechst. The movie starts 68 min before anaphase onset. Note that nuclear envelope (NE) aberrations manifest upon NE reformation after mitosis and persist.

https://elifesciences.org/articles/63614#video1

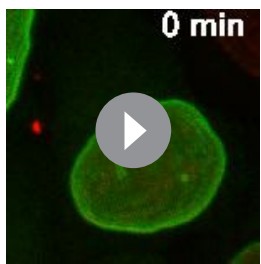

**Video 2.** Time-lapse imaging of a LAP2β-GFP expressing cell progressing through mitosis. Expression of LAP2β-GFP was induced 24 hr prior to imaging. DNA was visualized by SirHoechst. The movie starts 68 min before anaphase onset. Note that the nuclear envelope (NE) rapidly adopts a smooth topology soon after NE reformation.
https://elifesciences.org/articles/63614#video2

## The CBR is required for induction of LAP1-induced NE aberrations

To test whether chromatin interaction of the CBR is required for the post-mitotic NE aberrations induced by LAP1B expression, we analyzed the nuclear shape of cells upon expression of either LAP1B-GFP or LAP1B(ΔCBR)-GFP 48 hr after transfection of the respective constructs. Strikingly, deletion of the CBR abolished the occurrence of NE aberrations (*Figure 2H*).

To pinpoint residues within the CBR that contribute to mitotic chromatin association, we mutated conserved Ser and Thr residues within the CBR (*Figure 2G*). With this approach, we identified two residues (S108 and T124) that if mutated to glutamate affected NE localization of the LAP1(98–136)-SPAG4-GFP fusion protein during interphase (*Figure 2E*), abolished its binding to chromatin in metaphase (*Figure 2E*) and reduced its affinity for DNA and mononucleosomes in vitro (*Figure 2—figure supplement 2C, D*). Although these Ser and Thr residues could in principle be subject to mitotic phosphorylation, we have so far been unable to confirm mitotic phosphorylation of these sites. Comparison of the diffusional mobility of full-length LAP1B-GFP with LAP1B(ΔCBR)-GFP and LAP1B-2E(S108E,T124E)-GFP demonstrated that both mutant constructs exhibited a higher mobility at the NE, highlighting the contribution of the CBR and the identified residues to the retention of LAP1B at the INM (*Figure 2—figure supplement 3A, B*). Importantly, when we introduced both mutations into the membrane-bound, nucleoplasmic domain of LAP1B in the construct LAP1(1-359-2E)-GFP, binding to mitotic chromatin and post-mitotic NE aberrations were fully abolished (*Figure 2F*). Taken together, this data shows that chromatin-binding contributes to the retention of LAP1 at the NE and identifies two residues (S108 and T124) that are essential for chromatin binding of LAP1B during mitosis. Importantly, this establishes a direct link between the CBR and post-mitotic NE aberrations.

## LAP1B-induced NE aberrations can be rescued by Torsin co-expression

LAP1 is an established activator of Torsin AAA+ ATPase family members, that is of Torsin1A (Tor1A) and Torsin1B (Tor1B) (*Brown et al., 2014*; *Sosa et al., 2014*). The persistent chromatin association of overexpressed LAP1B during mitosis prompted us to examine whether restoring a normal LAP1B to Torsin ratio by co-overexpression of these proteins would rescue NE shape defects. We transiently expressed LAP1B-GFP alone or together with either Tor1A or Tor1B, both C-terminally tagged with an HA-Strep tag, and analyzed the percentage of cells containing NE aberrations two days after transfection. For quantification, we binned cells into a window of comparable fluorescence intensities to exclude an influence of varying LAP1B expression levels. Indeed, co-expression of either Torsin led to a striking decrease in NE aberrations (*Figure 3A, B, C*). In the following, we continued our work with Tor1B, since it had been previously demonstrated that its in vitro ATPase activity can be more potently stimulated by LAP1 than that of Tor1A (*Zhao et al., 2013*), indicating that Tor1B could be the more potent enzyme.

To test whether the observed rescue depends on the ATPase activity of Tor1B, we utilized a well-characterized Glu to Gln mutation in the Walker B motif that prevents ATP hydrolysis but not ATP binding, and can be used to lock AAA+ ATPases on their substrates (*Snider et al., 2008*; *Weibezahn et al., 2003*). In the case of Torsins, the ATPase-deficient E to Q mutant traps Torsins on its activators LAP1 and LULL1 (an ER-resident LAP1 paralog), while substrates of Torsins have not been identified (*Jungwirth et al., 2010*; *Rose et al., 2014*; *Zhao et al., 2013*). In contrast to wild-type Torsins, Tor1B(E178Q)-HASt overexpression failed to rescue the NE aberrations caused by LAP1B-GFP, indicating that the enzymatic activity of Torsins is needed to prevent the LAP1B-induced phenotypes and not the LAP1B to Torsin ratio alone (*Figure 3A, B, C*). As expected, NE aberrations triggered by expression of LAP1B(1-359)-GFP, which lacks the luminal domain required

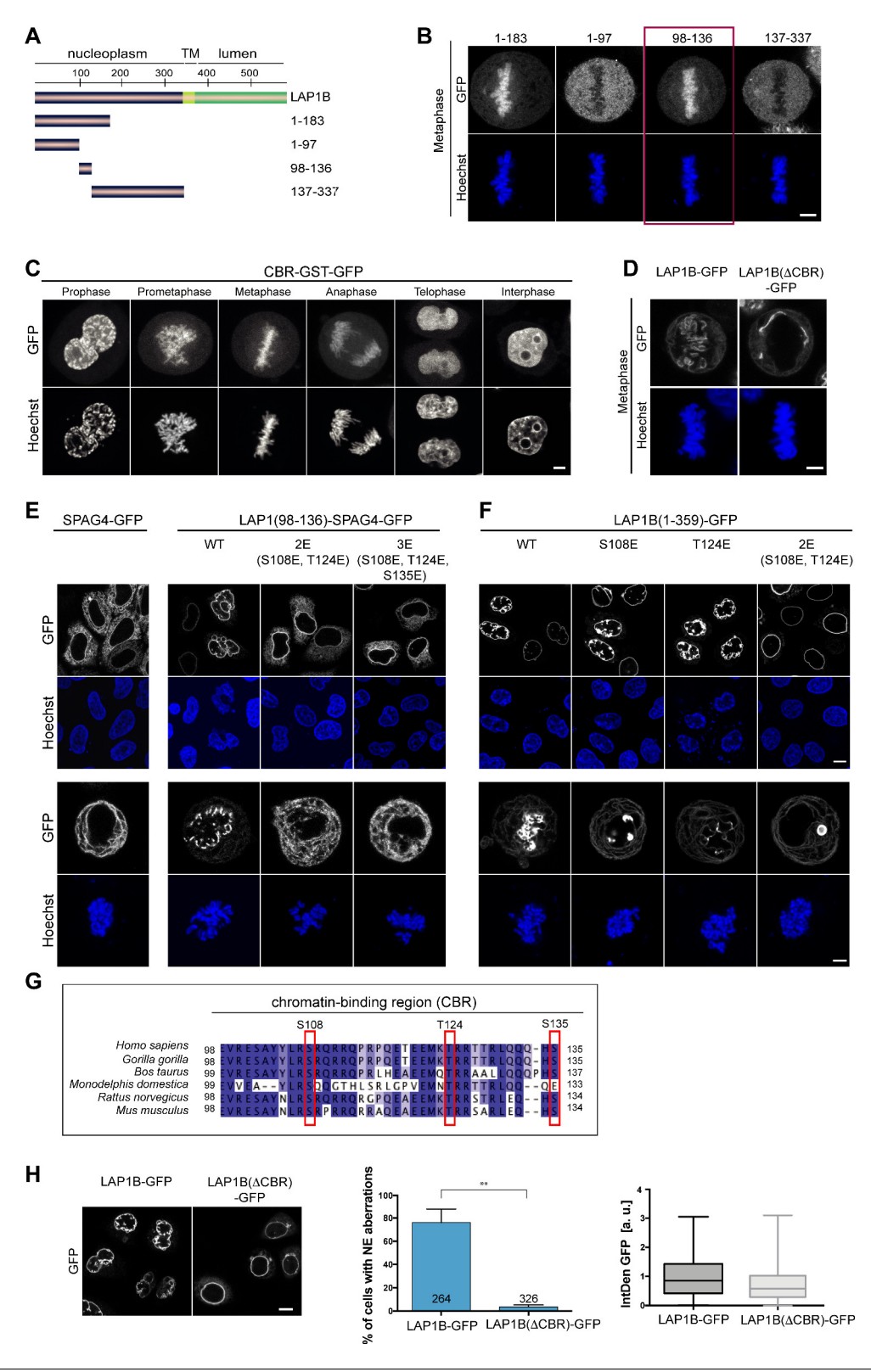

**Figure 2.** The nucleoplasmic domain of LAP1 contains a central chromatin-binding region (CBR) that can confer chromatin association during mitosis.
(A) Scheme depicting the generated fragments of the nucleoplasmic domain of LAP1B. (B) Metaphase localization of the depicted LAP1 fragments,
transiently expressed in synchronized HeLa cells. Maximum intensity z-projections (3 × 1 µm). Scale bar, 5 µm. (C) Localization of LAP1(CBR)-GST-GFP
during mitosis in HeLa cells. Scale bar, 5 µm. (D) LAP1B-GFP and LAP1B(ΔCBR)-GFP localization during metaphase. Scale bar, 5 µm. (E) Localization of
*Figure 2 continued on next page*

Figure 2 continued

wild-type SPAG4-GFP and (F) LAP1B(98-136)-SPAG4-GFP derivatives in interphase and prometaphase cells (arrested with nocodazole for 3 hr) 48 hr after transfection. Scale bars, 10 µm (upper panels) and 5 µm (lower panels). (G) Alignment of sequences of the CBR of LAP1B from the indicated mammalian species. The three mutated residues are boxed in red. (H) Left: Interphase HeLa cells expressing LAP1B-GFP or LAP1B(ΔCBR)-GFP for 48 hr. Scale bar, 10 µm. Middle and right: Percentage of nuclei with nuclear envelope (NE) aberrations, quantified as in *Figure 3*, in cells expressing LAP1B-GFP or LAP1B(ΔCBR)-GFP at similar levels, based on the integrated GFP density (IntDen GFP) (N = 3, n > 264, mean +/- SEM).

The online version of this article includes the following figure supplement(s) for figure 2:

**Figure supplement 1.** Delineation of the lamina-binding regions in LAP1B.
**Figure supplement 2.** Delineation of the chromatin-binding region (CBR) of LAP1B.
**Figure supplement 3.** The chromatin-binding region (CBR) of LAP1 contributes to its retention at the inner nuclear membrane.

for Torsin activation, were not rescued by Tor1B-HASt overexpression (*Figure 3A, B, C*). Similarly, Tor1B co-expression did not ameliorate NE aberrations caused by expression of LAP1B(R563G) (*Figure 3E, F*), a mutant inactivating the Arg-finger needed for stimulation of Torsin ATPase activity (*Brown et al., 2014*; *Sosa et al., 2014*). Another non-functional LAP1B mutant, E482A, known to cause severe dystonia, cerebellar atrophy and cardiomyopathy (*Dorboz et al., 2014*), also induced NE aberrations, which were not prevented by Torsin co-expression. Thus, Torsins can only act on LAP1 derivatives that are competent for the activation of its ATPase activity.

We also examined the influence of the ER-localized Torsin activator LULL1 as it had been suggested to regulate access of Torsins to the INM (*Goodchild et al., 2015*; *Vander Heyden et al., 2009*). However, expression of LULL1 did not improve the NE shape of LAP1B-GFP expressing cells (*Figure 3—figure supplement 1*). Taken together, the correct LAP1B to Torsin ratio and the functional interplay between LAP1B and Torsins ensure an unperturbed post-mitotic nuclear morphology.

## Torsins influence the diffusional mobility of LAP1 at the NE

Given that Torsins are expressed in postmitotic, differentiated cells, we next set out to examine whether Torsins influence LAP1 dynamics at the NE during interphase.

Firstly, we compared the mobility of the Torsin activation-deficient mutant R563G and the disease-causing mutant E482A to wild-type LAP1B by FRAP. The E482A mutation has been suggested to impair folding of the luminal domain (*Demircioglu et al., 2016*). Since LAP1B is so strongly immobilized at the INM by binding to nuclear lamins and to chromatin, it was not surprising that no mobility changes were detected (*Figure 4A*). However, upon depletion of laminA/C and laminB1, it became apparent that both mutants were less mobile compared to the wild-type protein (*Figure 4B*). Secondly, to more directly test whether Torsins might influence the retention of LAP1 at the INM, we measured the mobility of LAP1B in Tor1B wild-type and Tor1B(E178Q)-expressing cells in the presence and absence of nuclear lamins (*Figure 4C, D*). The presence of the dominant-negative Torsin indeed significantly reduced the mobility of LAP1B-GFP in laminA/C and B1-depleted cells. Thirdly, to address how chromatin interaction by the CBR of LAP1B influences this behavior, we exploited the identified 2E mutant that reduces the affinity of the CBR for chromatin (*Figure 2—figure supplement 2C, D*). As expected, the LAP1-2E mutant was more mobile, and its mobility seemed less reduced by Tor1B(E178Q) expression.

## LAP1 mutants unable to activate Torsins cause binucleation that depends on LAP1–chromatin interaction

We have previously shown that persistent membrane–chromatin interactions during mitosis leads to severe post-mitotic NE morphology defects, and, in addition, to a significant increase in chromosome segregation defects, eventually resulting in cell division failure and an increase in the number of binucleated cells (*Champion et al., 2019*). To elucidate whether persistent LAP1–chromatin interactions would mimic these phenotypes, we examined binucleation upon expression of LAP1B-GFP and its derivatives deficient in Torsin activation. To quantify binucleated cells, we visualized the cytoplasmic contours by tubulin immunofluorescence and counterstained nuclei with Hoechst (*Figure 5A, B, C*). Compared to the parental HeLa cell line, expression of LAP1B-GFP induced a two-fold increase in binucleated cells. At a similar level, both LAP1B(1-359)-GFP and LAP1B(R563G)-

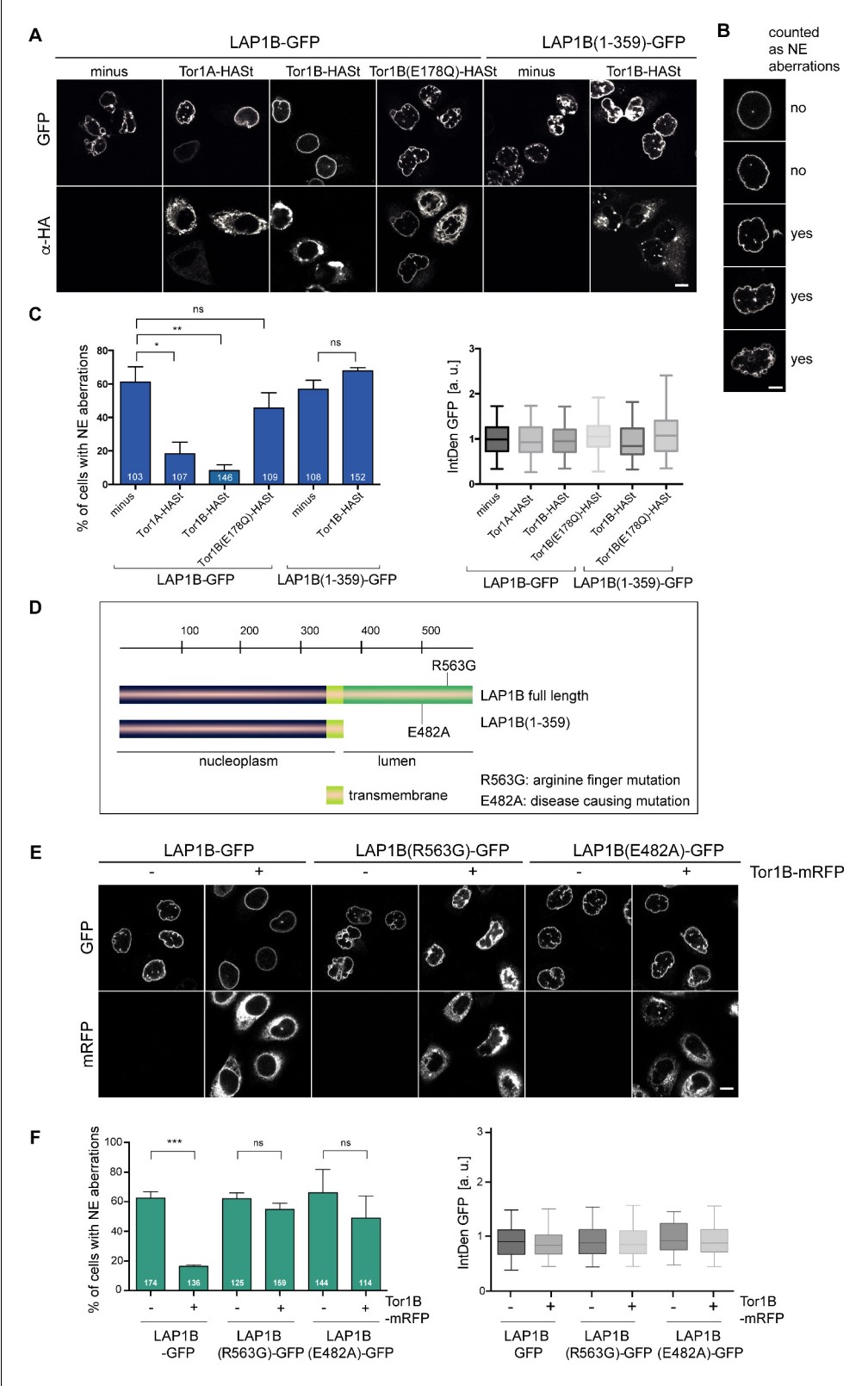

**Figure 3.** LAP1B-induced nuclear envelope (NE) aberrations can be rescued by co-expression of Torsins. (**A**) HeLa cells were transiently transfected with constructs encoding for LAP1B-GFP or LAP1B(1-359), either alone or together with the depicted Torsin constructs. After 48 hr, cells were fixed, subjected to immunostaining of Torsins using an anti-HA antibody and analyzed by confocal microscopy. Scale bar, 10 μm. (**B**) Representative images of pre-defined classes of cell nuclei used for the classification of NE aberration phenotypes. Based on nuclear shape, cells were assigned into one of

*Figure 3 continued on next page*

*Figure 3 continued*

the five shown categories for the quantification in C. The two upper classes were regarded as normal (no; no NE aberration), and the three lower classes as aberrant (yes). Scale bar, 5 µm. (**C**) The percentage of cells with NE aberrations was quantified by assigning cells into one of five predefined phenotypic classes shown in B. (left; N = 3, n > 103, mean +/- SEM). Only cells with similar LAP1B-GFP expression levels, based on quantification of the integrated GFP density per cell (IntDen GFP, normalized to the LAP1B minus Torsin control; right) were considered. (**D**) Depiction of the mutant LAP1B constructs; R563G: Torsin activation deficient-mutant; E482A, disease-causing mutant. (**E**) HeLa cells were transiently transfected with constructs encoding for the indicated LAP1B-GFP derivatives, either alone or together with Tor1B-mRFP. After 48 hr, cells were fixed and analyzed by confocal microscopy. Scale bar, 10 µm. (**F**) The fraction of cells with NE aberrations was quantified as in C (N = 3, n > 114, mean +/- SEM).

The online version of this article includes the following figure supplement(s) for figure 3:

**Figure supplement 1.** Nuclear envelope (NE) aberrations induced by LAP1B cannot be rescued by LULL1 co-expression.

GFP led to a twofold further rise in the number of binucleated cells compared to cells expressing LAP1B wild-type. This number was even higher for the disease-causing mutant LAP1B(E482A)-GFP. Thus, the E482A mutant seems to be a very potent dominant-negative if expressed.

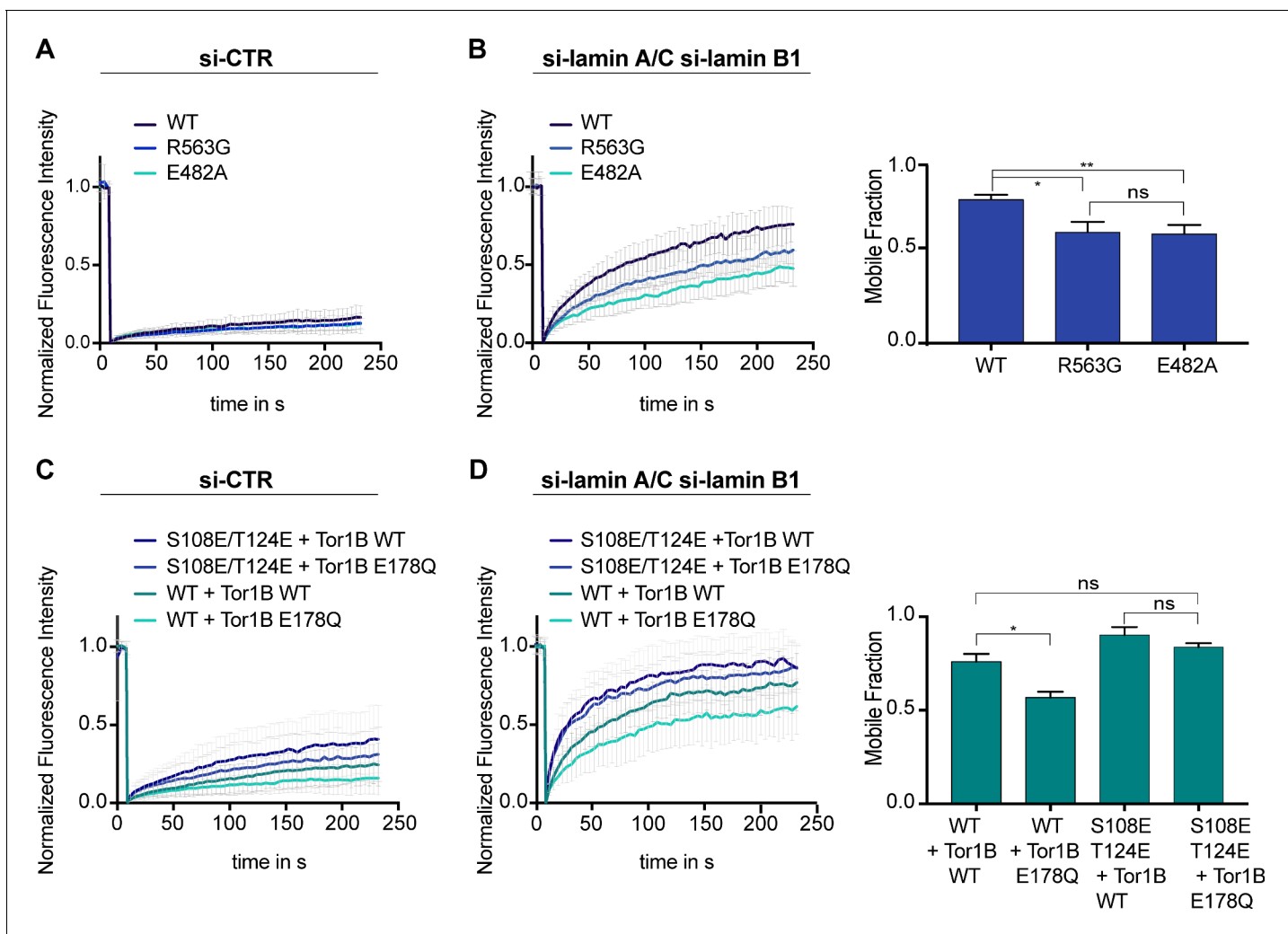

**Figure 4.** Mobility of LAP1B at the nuclear envelope (NE) of interphase cells is influenced by Torsins. (**A**) FRAP analysis of HeLa cells expressing either wildtype LAP1B-GFP or the indicated mutant variants and treated with a control siRNA (N = 5, n > 18, mean +/- SD). (**B**) FRAP analysis as in B after RNAi-mediated co-depletion lamin A/C and lamin B1 (N = 5, n > 21, mean +/- SD). Corresponding mobile fractions (mean +/- SEM, *p<0.05). (**C**) FRAP analysis of LAP1B-GFP and the chromatin-binding deficient variants in cells expressing either Tor1B wild-type or the dominant-negative Tor1B(E178Q) mutant both tagged with mRFP, and treated with a control siRNA (N = 4, n > 18, mean +/- SD). (**D**) FRAP analysis in lamin-depleted cells as in C (N = 4, n > 25, mean +/- SD). Corresponding mobile fractions (mean +/-, *p<0.05).

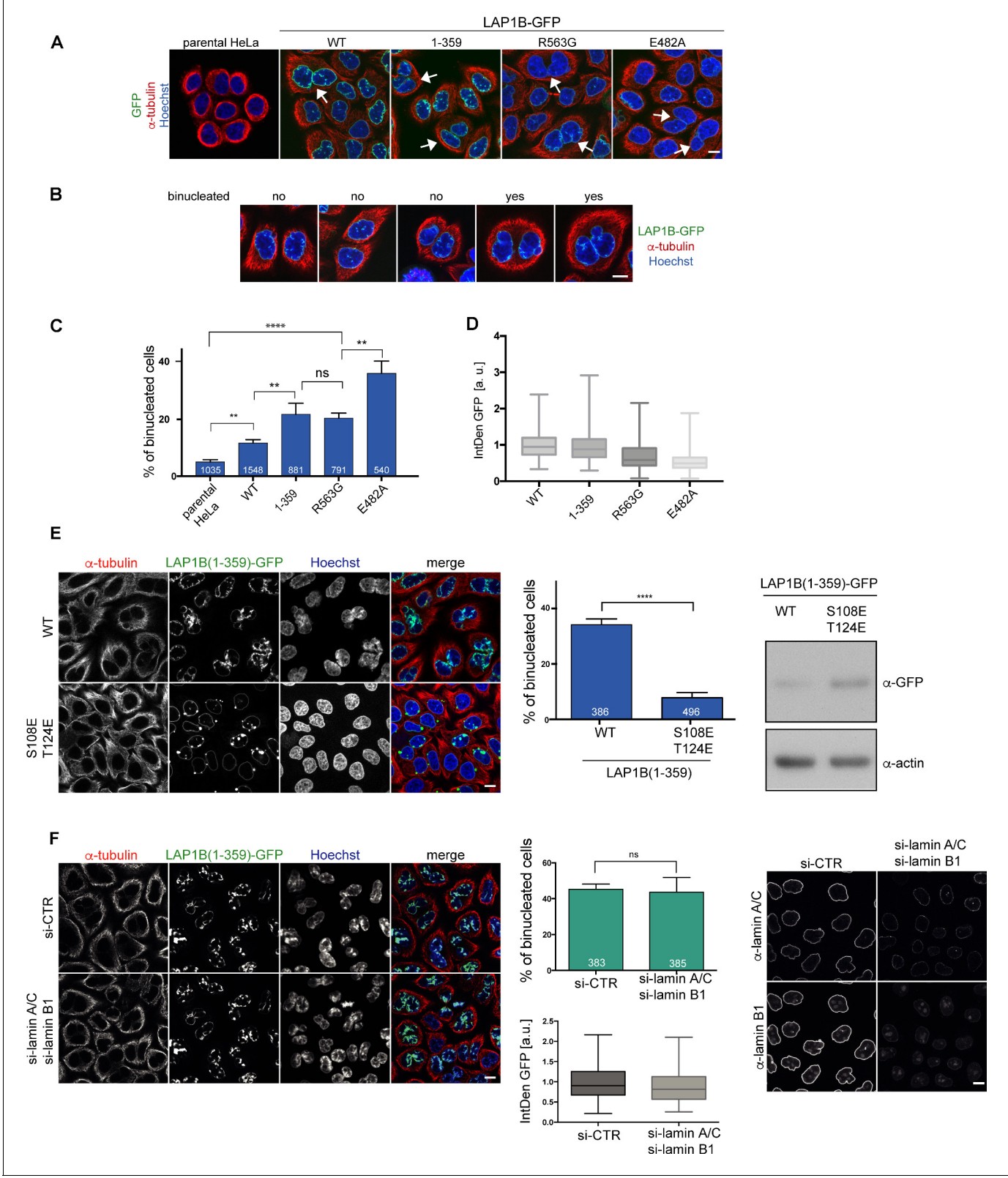

**Figure 5.** LAP1B mutants deficient in Torsin activation lead to increased binucleation dependent on chromatin interaction of LAP1. (**A**) Expression of LAP1B(1-359)-GFP was induced with 0.01 µg/ml tetracycline and expression of LAP1B-GFP wild-type, LAP1(R563G)-GFP and LAP1(E482A)-GFP with 1 µg/ml tetracycline for 48 hr. The parental HeLa cell line was used as a negative control. Then, cells were fixed, immunostained for tubulin and analyzed by confocal microscopy. Scale bar, 10 µm. White arrows denote binucleated cells. (**B**) Images representing the categories used for classification of
*Figure 5 continued on next page*

*Figure 5 continued*

binucleation. (C) Quantification of the fraction of binucleated cells (N = 3, n > 540, mean +/- SEM). (D) The integrated density (IntDen) of the GFP signal was measured and normalized to LAP1B-GFP. (E) Binucleation caused by LAP1B is prevented by point mutations that abolish chromatin-binding. Expression of LAP1B(1-359)-GFP or LAP1B(1-359-2E)-GFP was induced with tetracycline (0.01 µg/ml and 1 µg/ml, respectively) in stable HeLa cell lines for 48 hr. Cells were immunostained for tubulin and counterstained with Hoechst. Binucleated cells were quantified (N = 3, n > 300, mean +/- SEM; ****p<0.0001). Expression levels of LAP1B(1-359)-GFP and LAP1B(1-359-2E)-GFP were compared by immunoblotting. (F) LAP1B(1-359)-GFP expression was induced for 48 hr as in E in either mock-treated or in lamin A/C and lamin B1-co-depleted cells (RNAi for 72 hr, induction of LAP1B constructs for the last 48 hr). Cells fixed and analyzed as in E. The number of binucleated cells was quantified (N = 3, n > 383, mean +/- SEM). LAP1B(1-359)-GFP expression levels were determined based on the integrated GFP intensity (IntDen). Lamin RNAi was controlled by immunofluorescence. Scale bars, 10 µm.

Next, we addressed the contribution of chromatin interaction of LAP1 to the formation of binucleated cells by comparing HeLa cell lines expressing either LAP1B(1-359)-GFP or the chromatin-binding-deficient 2E (S108E, T124E) mutant, comparing cells at a high expression level by using an elevated tetracycline concentration for induction of the constructs. Whereas LAP1B(1-359)-GFP caused binucleation in nearly 40% of cells under these conditions, LAP1B(1-359-2E)-GFP did not (*Figure 5E, F*), confirming that binucleation is caused by the LAP1B-chromatin interaction. As LAP1B also interacts with lamin A/C and lamin B1, we further tested whether LAP1–lamina interactions contribute to binucleation. As RNAi-mediated depletion of lamin A/C and laminB1 in cells expressing LAP1B(1-359)-GFP did not reduce the number of binucleated cells (*Figure 5G, H*), we conclude that the cell division defects are independent of nuclear lamins, and thus solely related to the aberrant binding of LAP1 to chromatin during mitosis.

## An ATPase-deficient Torsin impairs the removal of endogenous LAP1 from mitotic chromatin and increases binucleation

To date, five human Torsins have been identified. Because these five members of the Torsin family likely function, at least in part, in a redundant manner, it is challenging to reach a sufficient downregulation of these enzymes by conventional protein depletion experiments. As dominant-negative approaches have been decisive in elucidating the cellular function of ATPases, we therefore decided to investigate the consequences of expression of the dominant-negative Tor1B(E178Q) mutant. Strikingly, when we visualized endogenous LAP1 by immunofluorescence in cells expressing Tor1B (E178Q)-GFP, we observed LAP1-positive patches localizing to (pro)-metaphase chromatin, indicating an impaired release of LAP1 from chromatin during mitotic entry (*Figure 6A*). In comparison, cells expressing wild-type Tor1B-GFP showed GFP-positive membrane patches in the peripheral mitotic ER. These patches are reminiscent of previously described ER membrane foci in TorsinB-expressing interphase cells (*Rose et al., 2014*).

Next, we compared the formation of binucleated cells upon induction of Tor1B(E178Q)-GFP or wild-type Tor1B-GFP. Remarkably, Tor1B(E178Q)-GFP expression led to a twofold higher number of binucleated cells as compared to cells expressing Tor1B-GFP wild-type (*Figure 6B*). Notably, no severe post-mitotic NE aberrations were observed in Tor1B(E178Q) expressing cells, which might be explained by fewer, weaker and/or less persistent chromatin interactions of endogenous LAP1 in anaphase compared to overexpressed LAP1. We know from our previous work that late mitosis is indeed critical for the manifestation of this phenotype (*Champion et al., 2019*). As binucleation is indicative for failed cytokinesis and commonly caused by chromosome segregation defects, we quantified DNA bridges and lagging chromosomes. This analysis revealed a significant increase in chromosome segregation defects (*Figure 6D*). To elucidate whether the observed increase in binucleated cells can be attributed to the failed removal of LAP1 from mitotic chromatin, we performed RNAi-mediated depletion of LAP1 in Tor1B(E178Q)-GFP expressing cells. Importantly, the number of binucleated cells was significantly reduced in LAP1-depleted cells (*Figure 6C*). As judged by the GFP signal, the Tor1B(E178Q)-GFP expression seemed slightly reduced in LAP1-depleted cells as compared to cells treated with control siRNA (*Figure 6C*; *Figure 6—figure supplement 1*), and we wished to exclude that unequal Tor1B(E178Q) expression levels led to the reduction of binucleation in LAP1-depleted cells. Therefore, we induced Tor1B(E178Q) with a lower tetracycline concentration in control cells. Even though these cells now expressed two-fold less Tor1B-E178Q compared to LAP1-depleted cells, binucleation was still significantly reduced upon downregulation of LAP1

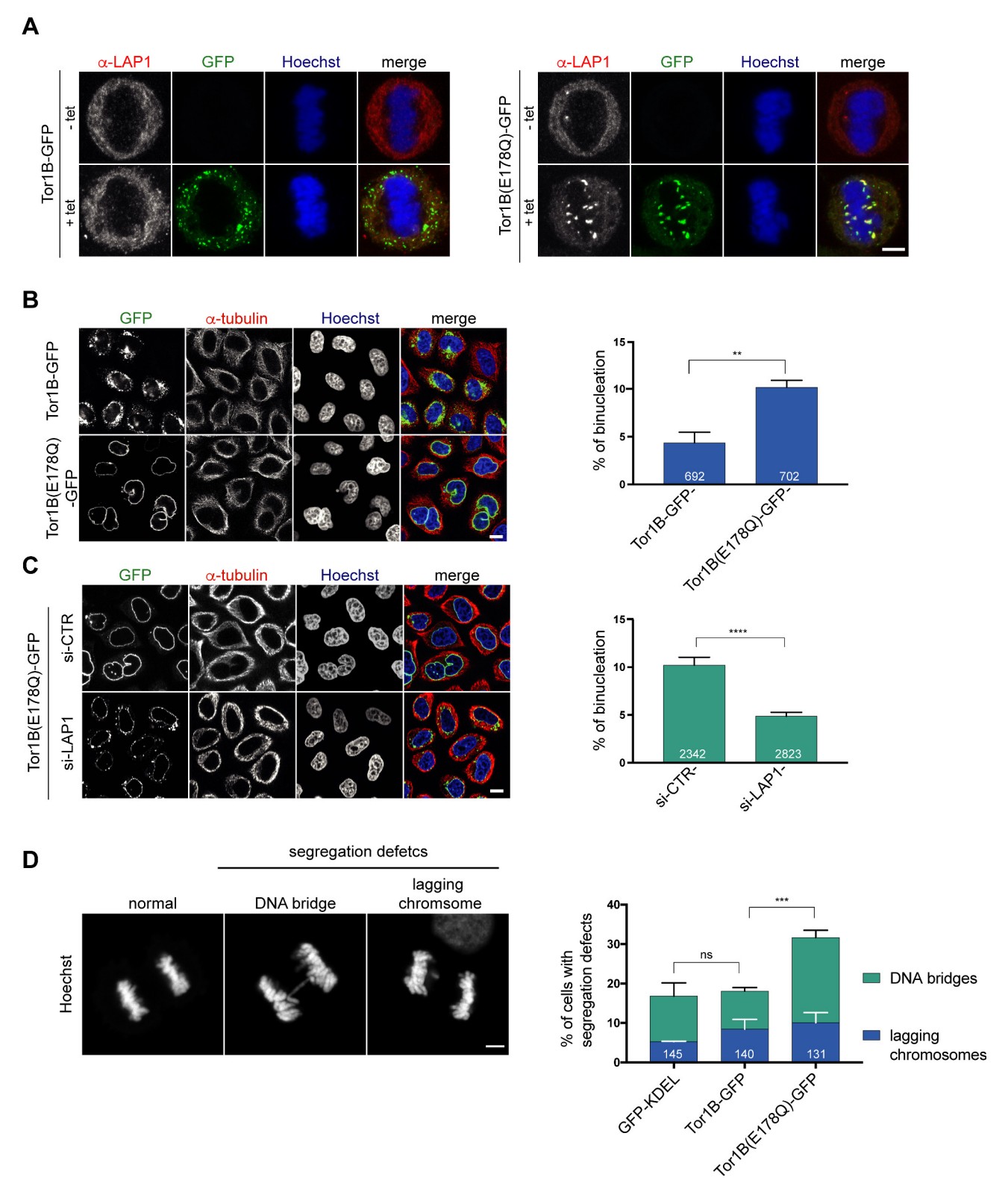

**Figure 6.** A dominant-negative, ATPase-deficient Torsin1B mutant increases binucleation in a LAP1-dependent manner. (**A**) Endogenous LAP1 remains associated with chromatin in metaphase upon overexpression of Tor1B(E178Q)-GFP. Representative maximum intensity projections (5 × 0.63 µm) of confocal images of metaphase HeLa cells upon induction of either Tor1B-GFP and Tor1B(E178Q)-GFP expression with 1 µg/ µl tetracycline for 48 hr. Localization of endogenous LAP1 was analyzed by immunofluorescence staining. Scale bar, 5 µm. (**B**) Confocal images of Tor1B-GFP and Tor1B

*Figure 6 continued on next page*

Figure 6 continued

(E178Q)-GFP-expressing cells stained with Hoechst and immunostained for tubulin. Scale bar, 10 μm. Quantification of binucleated cells (N = 3, n > 835, mean +/- SD). (C) Expression of Tor1B(E178Q)-GFP was induced for 48 hr in LAP1-depleted or mock-treated cells. Binucleation was analyzed by confocal microscopy and quantified (N = 3, n > 1915, mean+/-SEM). Scale bar, 10 μm. (D) Quantification of chromosome segregation defects (DNA bridges and lagging chromosomes) in fixed GFP-KDEL, Tor1B-GFP, and Tor1B(E178Q)-GFP expressing anaphase cells after 24 hr of induction with tetracycline. Segregation errors were manually quantified (N = 3, n > 130, mean+/-SEM).

The online version of this article includes the following figure supplement(s) for figure 6:

**Figure supplement 1.** Expression of a dominant-negative Torsin1B variant leads to an increase in binucleated cells in a LAP1-dependent manner.
**Figure supplement 2.** Downregulation of LULL1 does not prevent Tor1B(E178Q)-induced binucleation.

(*Figure 6—figure supplement 1A, B, C*). For comparison, we also depleted the ER-resident Torsin-activator LULL1 in Tor1B(E178Q)-GFP expressing cells, but could not observe a reduction in binucleated cells (*Figure 6—figure supplement 2A, B, D*). LULL1 depletion also did not prevent the localization of LAP1 to mitotic chromatin upon Tor1B(E178Q) expression (*Figure 6—figure supplement 2C, D*). Taken together, the dominant-negative Tor1B variant induces the persistence of LAP1-positive membrane patches on (pro)-metaphase chromatin, chromosome segregation defects and binucleation in a LAP1-dependent manner. Collectively, our findings indicate that Torsin function is important for mitotic fidelity by assuring normal LAP1 behavior during mitosis.

## Discussion

During NE breakdown, membrane proteins of the INM are released from chromatin and the nuclear lamina to allow for the spatial separation of membranes from chromatin. Our work uncovers that the abundant and ubiquitously expressed INM protein LAP1 is tightly anchored at the INM of interphase cells by association with both chromatin and the nuclear lamina. Unexpectedly, we found that LAP1 remains stuck on chromatin in mitosis if either the stoichiometry of LAP1 to Torsins is unbalanced or Torsin functionality is compromised. These findings are intriguing for two reasons. Firstly, they indicate that Torsins, residing in the perinuclear space, influence LAP1's ability to bind chromatin at the other side of the membrane. Secondly, reminiscent of some of the phenotypes that we have previously observed using a synthetic membrane-chromatin tether (*Champion et al., 2019*), the failure in release of LAP1 from mitotic chromatin is associated with chromosome segregation defects, binucleation and postmitotic NE aberrations.

### The Torsin-LAP1 interplay

Based on studies in human cells, mice and *Drosophila* diverse roles of Torsins have been suggested, including functions in ER homeostasis, secretion, lipid metabolism, NPC biogenesis, and nuclear export of large particles like mRNPs or HSV1 capsids by a vesicular transport process across the NE (*Chen et al., 2010*; *Grillet et al., 2016*; *Jokhi et al., 2013*; *Maric et al., 2011*; *Nery et al., 2011*; *Pappas et al., 2018*; *Rampello et al., 2020*; *Rampello et al., 2020*; *Shin et al., 2019*; *Turner et al., 2015*). However, while all these suggested functions are supported by the reported cellular phenotypes, biochemical and genetic data, it is currently unclear what are direct or indirect consequences of Torsin function since the molecular mechanism of Torsin action has remained elusive. All attempts to identify bona fide substrates of these unsual, ER-luminal AAA+ ATPases have so far been unsuccessful. It has therefore even been speculated that the established Torsin activators LAP1 and LULL1 could act simultaneously as activators and substrates of these mysterious ATPases (*Cascalho et al., 2017*; *Rose et al., 2015*), especially since LAP1 and LULL1 are highly enriched on the ATP-bound form of some Torsins, as expected for substrates of AAA+ ATPases.

So, how do our findings relate to the current understanding of the molecular function of Torsins? Our experiments suggest that the interaction of Torsins with the C-terminal, luminal domain of LAP1 in the ER lumen influences the chromatin-binding activity of LAP1's nucleoplasmic domain on the other side of the membrane. This implies that LAP1 might not merely behave as an activator of Torsins, but could in turn itself be a downstream effector or 'substrate' influenced by the activity of the ATPases. Two observations indicate that this reverse crosstalk from Torsin to LAP1 is indeed linked to the enzymatic activity of the ATPase: Firstly, LAP1B-induced NE aberrations can be rescued by expression of Tor1B wild-type but not by an ATPase inactive mutant. Secondly, LAP1 derivatives

that are deficient in Torsin ATPase activation execute a stronger dominant-negative effect on cell division than wild-type LAP1.

Clearly, even if considering LAP1 a 'substrate' of Torsins ATPases, their interplay must differ from the 'classical' enzyme-substrate mechanism typical of AAA+ ATPases that thread protein substrates through the central pore of a usually hexameric ATPase rings (*Hanson and Whiteheart, 2005*; *Olivares et al., 2016*). Although Torsins show sequence similarity to bacterial ClpX ATPases that serve as the paradigm of substrate-threading AAA+ ATPases (*Olivares et al., 2016*), they seem to lack the characteristic pore loops involved in substrate gripping during the translocation of an unfolded protein chain through the central opening of the ring (*Demircioglu et al., 2016*). Furthermore, the luminal domains of LAP1 and LULL1 are structural mimics of the ATPase fold. These domains bind laterally to the ATPase domain of Torsins to activate ATP hydrolysis through their arginine finger (*Brown et al., 2014*; *Sosa et al., 2014*), inconsistent with LAP1 being a substrate of a threading mechanism. Consistently, Torsins have not been found to unfold the luminal domains of LAP1 and LULL1 in vitro (*Zhao et al., 2013*). Thus, the interplay of Torsin and LAP1 must work differently.

From a conceptual point of view, we can envisage several scenarios as to how Torsin ATPase activity could influence chromatin interaction of LAP1. One possibility is that activation of Torsins transduces mechanical forces across the membranes, thereby weakening the interaction of LAP1 with chromatin, and thereby facilitating its removal from chromatin. This might entail a conformational rearrangement of the luminal domain of LAP1 relative to the ATPase domain of Torsins induced by ATP hydrolysis, and propagated to the other side of the membrane either in form of a Torsin-induced pulling on LAP1 or a rotational movement of LAP1 along its longitudinal axis in the plane of the membrane. Another plausible scenario is that Torsin ATPase activation leads to the disassembly of LAP1 dimers or higher order oligomers by affecting their lateral association, thereby influencing the avidity of the LAP1–chromatin interaction.

Based on structural studies that revealed an AAA+ fold mimicry of the luminal domains of LAP1 and LULL1, it has originally been assumed that they form mixed heterohexameric rings with Torsins, with support from in vitro assembly experiments (*Brown et al., 2014*; *Sosa et al., 2014*). Such a heterohexameric configuration with 3 LAP1 and 3 Torsin molecules per hexamer would readily explain how an ATPase-deficient Torsin mutant such as Tor1B(E178Q) could increase the affinity of LAP1 for chromatin due to avidity effects caused by the entrapment of LAP1 in these ATP-stabilized ring structures. However, it was subsequently noted that the surfaces of the luminal domains of LAP1 and LULL1 are only well conserved on their catalytic interface with Torsins but not on the potential non-catalytic interface, indicating the lack of evolutionary pressure to maintain this interface and shedding some doubt on the suggested hetero-hexameric organization (*Chase et al., 2017*; *Demircioglu et al., 2016*). In contrast, Torsins display surface conservation on both their catalytic and non-catalytic interfaces, consistent with the possibility that Torsins could form homo-oligomers. Indeed, Torsins can assemble into ATP-dependent, homo-oligomeric structures in vitro, adopting the configuration of a spiral or potentially a lock-washer (*Chase et al., 2017*; *Demircioglu et al., 2016*). Such configuration would provide only one accessible interface for binding of LAP1 or LULL1 to promote oligomer disassembly upon ATPase activation. It remains to be seen which higher order configuration of Torsins exist in living cells, also in light of the view that LAP1 and LULL1 are over-stoichiometric to Torsins (*Itzhak et al., 2016*). Such knowledge will be key to inform rational models of how Torsins influence chromatin interaction of LAP1.

## LAP1 interaction with chromatin and the nuclear lamina

LAP1B possesses an extremely low diffusional mobility in comparison to other INM proteins (*Zuleger et al., 2011*). Its intrinsically disordered nucleoplasmic domain engages in interaction with at least three binding partners that contribute to its strong retention at the INM, namely lamin A/C, lamin B1, and chromatin. This raises the question of whether the strong immobilization of LAP1 at the NE might bear an additional function beyond retaining the protein at the INM. Potential roles include the mechanical coupling of the INM to chromatin and the nuclear lamina or a function in chromatin positioning at the nuclear periphery, for example in heterochromatin organization. INM proteins together with nuclear lamins are known to serve as heterochromatin tethers that promote the positioning of transcriptionally silenced heterochromatin domains underneath the NE of differentiated cells (*Guelen et al., 2008*; *Solovei et al., 2013*). Considering the organization of its

nucleoplasmic domain, LAP1 seems ideally suited to bridge lamina-chromatin interactions. Notably, LAP1 is developmentally essential as LAP1 knockout mice exhibit perinatal lethality (*Kim et al., 2010*). Recently, loss of LAP1 expression in human patients caused by a mutation leading to an in-frame termination codon in LAP1 has been associated with an early onset multisystemic nuclear envelopathy manifesting in severe neurological and developmental deficiencies, resulting in early lethality of the patients (*Fichtman et al., 2019*). These severe defects are most likely caused by a lack of Torsin activation at the NE by lack of LAP1. However, it might also be worth considering whether LAP1 might have a chromatin-associated function that is needed for proper development.

Our study primarily focused on the long isoform LAP1B. In comparison, LAP1C is predicted to possess a shortened CBR. When overexpressed, LAP1C is also capable of binding to chromatin during mitosis, albeit to a lesser extent than LAP1B. Interestingly, the relative expression of LAP1B and LAP1C differs between tissues and changes during development (*Santos et al., 2014*; *Shin et al., 2014*). While LAP1C is the predominant isoform in undifferentiated cells, the relative levels of LAP1B strongly increase during differentiation and, interestingly, upon induced cell cycle exit of cultured somatic cells (*Santos et al., 2015*). In light of our observation that LAP1B is more prone to stick to mitotic chromatin than LAP1C and induces mitotic errors when present at increased levels in rapidly dividing cells, the physiological upregulation of LAP1B in correlation with cell cycle exit seems ideally suited to prevent potentially fatal mitotic defects caused by LAP1B.

## The gist of the matter – mitosis or interphase?

Although we present evidence that Torsins influence the interaction of LAP1 with chromatin, we do not necessarily assume that Torsins have evolved to liberate LAP1 from chromatin for open mitosis or that they possess a function dedicated to mitosis. Given that primary dystonia caused by mutations of Tor1A or LAP1 seems to arise through a dysfunction of post-mitotic neurons, the function of Torsins is probably essential for the homeostasis of interphase cells. We rather consider it likely that mitosis is perturbed as a consequence of a perturbed LAP1-Torsin interplay during interphase that manifests in the subsequent mitosis.

Both the expression of the Tor1B ATPase-deficient E178Q mutant or excess LAP1 impaired the full removal of LAP1 from chromatin in mitosis, and is associated with chromosome segregation errors, increased binucleation of cells and post-mitotic NE aberrations. When we weakened chromatin association of LAP1B by mutation of two residues in the CBR to phospho-mimetic residues, these phenotypes disappeared, suggesting that the prolonged chromatin association of LAP1 during mitosis is the underlying reason. Dissociation of membrane chromatin contacts is currently assumed to be primarily controlled by protein phosphorylation. While our own investigations of mitotic phosphorylation of these sites remained inconclusive, one of these residues, that is Ser135, has indeed been detected as mitotically phosphorylated in large scale phosphoproteomics (*Dephoure et al., 2008*). If chromatin interaction of LAP1 were normally controlled by (mitotic) kinases, then the perturbed Torsin-LAP1 interplay could also affect the access of kinases to the nucleoplasmic domain of LAP1, for example by a failure in dissolving interactions of LAP1 with either itself or other partners. Interestingly, the nucleoplasmic domain of LAP1 is known to interact with protein phosphatase 1, the persistent binding of which could counteract the action of (mitotic) kinases (*Santos et al., 2013*). Finally, mitotic changes in chromatin may also promote LAP1 dissociation during mitosis, and these changes might be affected by perturbation of the LAP1-Torsin axis.

Both binucleation and aneuploidy may contribute to or even drive tumorigenesis (*Chow et al., 2012*; *Levine and Holland, 2018*; *Tanaka et al., 2018*). Because Torsins likely function in a redundant manner, it is highly improbable that Torsin function can be completely abolished by mutations in most tissues. For LAP1, in contrast, mutations that specifically impair its Torsin binding or activation capability, present a risk of causing mitotic errors. Remarkably, mutation of LAP1's arginine finger was identified as a somatic mutation in melanoma (*Krauthammer et al., 2012*) and colon cancer (*The Cancer Genome Atlas Network, 2012*). However, whether this mutation genuinely contributes to tumorigenesis remains to be clarified. Based on our findings, one could expect aneuploidy to develop over time if differentiated cells with high LAP1B expression levels were forced to reenter the cell cycle for instance by oncogenic driver mutations.

## Materials and methods

A list of materials is provided in Appendix 1-Key resource table 1.

### Molecular cloning

The LAP1B, LULL1, Tor1A, and Tor1B coding regions were amplified from HeLa cDNA and cloned into the vector pcDNA5.1-GFP FRT/TO (cytomegalovirus [CMV] promoter; Invitrogen) for generation of stable, tetracycline-inducible cell lines. The respective constructs for LAP1B (isoform 3) and Tor1B pcDNA5.1 FRT/TO served as PCR or mutagenesis templates, yielding the following derivatives: LAP1C, LAP1B(1-359), LAP1B(Δ98–136), LAP1B(73-end), LAP1B(184-end), LAP1B(R563E), LAP1B (E482A), LAP1B(S108E,T124E), LAP1B(1-359)(S108E,T124E), LAP1B(S108E,T124E, R563G), LAP1(98–136), LAP1(98–136)(S108E,T124E), and Tor1B(E178Q). Fragments of the nucleoplasmic domain of LAP1 were subcloned into the mammalian expression vector pK7-GST-GFP (*Ungricht et al., 2015a*) and/or pK7-GST-GFP-NLS(SV40) (*Erkmann et al., 2005*) or the bacterial expression vector pQE60-zz-6xhis (*Kutay et al., 1997*), encoding for two z domains derived from the immunoglobulin-binding domain of *Staphylococcus aureus* protein A. For the domain swap experiment with SPAG4, the chromatin binding region of LAP1 (aa 98–136) and mutants thereof were cloned into pEGFP-N3-SPAG4 (*Turgay et al., 2010*). The emerin and LEM2 coding regions were amplified from HeLa cells derived cDNA and cloned into pEGFP-N3 (Clontech). pEGFP-N3-derived plasmids encoding for LAP2β (*Ungricht et al., 2015a*), SUN1 and SUN2 (*Turgay et al., 2010*) have been described before. Point mutations were introduced by using QuikChange site-directed mutagenesis.

### Antibodies

The following commercial antibodies were used in this study: anti-β-actin (mouse; Sigma, A1978; RRID:AB_476692), anti-emerin (rabbit, abcam Ab40688, RRID:AB_2100059), anti-HA (mouse, Covance, MMS-101P, RRID:AB_2314672), anti-lamin A/C (mouse, ImmuQuest (IQ332 RRID:AB_10660272)), anti-lamin B1 (rabbit, abcam ab16048, RRID:AB_443298), anti-lamin B2 (rabbit, abcam (ab151735) RRID:AB_2827514), anti-LAP1 (rabbit, abcam, ab86307 RRID:AB_2206124), anti-LBR (rabbit; Abnova, PAB15583; RRID:AB_10696691), mAB414 (mouse; Abcam, ab 24609; RRID:AB_448181), anti-H3 (rabbit, acam ab1791, RRID:AB_302613), anti-Tor1A (rabbit, Abexxa, abx001683), anti-Tor1B (rabbit, antibodies-online, ABIN1860834), anti-HSP60 (rabbit, abcam, ab45134, RRID:AB_733033), anti-α-tubulin (mouse, Sigma, T5168, RRID:AB_477579).

Antibodies were raised in rabbits against purified, recombinant human LULL1. Antibodies against human SUN1 (*Sosa et al., 2012*), human SUN2 (*Turgay et al., 2010*) and anti-GFP (*Turgay et al., 2014*) have been previously described.

### Generation of LMN A/C knockout cells

LMNA/C KO cell lines were made using the CRISPR/Cas9 system. The guide RNA (gRNA) was designed using the CRISPR design web tools (http://www.e-crisp.org/E-CRISP/) and targeted exon 1 of the *LMNA/C* gene. The annealed guide RNA-encoding DNA oligos (5'-CACCGGG TGGCGCGCCGCTGGGACG-3' and 5'-AAACCGTCCCAGCGGCGCGCCACCC-3') were ligated into the pC2P vector (*Welte et al., 2019*), which encodes for hCas9 and possesses a puromycin resistance cassette. Cells were transfected with the resulting pC2P-gLMNA/C vector and selected with puromycin for 3 days. Surviving clones were expanded and screened for mutations in the *LMNA/C* gene by immunoblotting and PCR. PCR products covering the edited genomic region were sequenced and analyzed for indel mutations using the tide web tool (*Brinkman et al., 2014*) and manual inspection of the sequencing chromatograms.

### Cell culture and synchronization of cells

HeLa FRT/TetR, HeLa K, HCT116, and HepG2 were kind gifts from T. Mayer (Department of Biology, Konstanz), B. Vogelstein (Johns Hopkins, Baltimore), D. Gerlich (Institute of Molecular Biotechnology, Vienna), and S. Werner (Institute of Molecular Health Science, ETH Zurich). These cell lines were not further authenticated after obtaining them from the indicated sources. All cell lines were tested negative for mycoplasma using PCR-based testing. None of these cell lines was included in the list of commonly misidentified cell lines maintained by International Cell Line Authentication Committee. Cells were cultivated in DMEM containing 10% (v/v) FBS and 100 µg/ml penicillin/streptomycin at

37°C with 5% $CO_2$ in a humidified incubator. Transient transfections were performed using X-treme-Gene (Roche) and jetPRIME (PolyPlus) transfection reagent. Stable tetracycline-inducible HeLa cell lines were generated by integration of the respective constructs into HeLa FRT/TetR cells.

Expression of the LAP1B-GFP, LAP1C-GFP, LAP2β-GFP, LAP1B(R563G)-GFP, LAP1B(E482A)-GFP, LAP1B(ΔCBR)-GFP, LAP1B(1-359)-GFP, LAP1B(S108E,T124E), Tor1B-GFP and Tor1B(E178Q)-GFP was induced with tetracycline as described in the figure legends. To synchronize HeLa cells in mitosis, a thymidine block was performed by adding 3 mM thymidine (Sigma-Aldrich) for 16 hr, followed by release of the cells from G1/S arrest with fresh medium for 10 hr. To arrest cells in prometaphase, 100 ng/ml nocodazole (Sigma-Aldrich) was added 8 hr after thymidine release.

Plasmid and siRNA transfections siRNA transfections were performed using INTERFERin (Polyplus) as transfection reagent. Cells were treated with 20 nM siRNA for 48 hr or 72 hr. The following siRNAs were purchased: si-CTR (AllStars negative control, Qiagen), si-lamin A/C (CUGGACUUCCAGAAGAACA, Microsynth), si-lamin B1 (UUCCGCCUCAGCCACUGGAAAU, Sigma), si-lamin B2 (ACAACUCGGACAAGGAUC, Microsynth), si-LAP1 (CUCACUAAGUUUCCUGAGUUA, Microsynth), si-LULL1 (CTGGTCCTGACTGTTCTGCTA, Microsynth).

## Immunofluorescence analysis

Cells grown on glass coverslips were washed with PBS and fixed for 10 min with either 4% PFA at RT or with methanol at - 20°C. After PFA fixation, cells were permeabilized with 0.1% Triton X-100 in PBS for 10 min. Then, cells were blocked with 2% (w/v) BSA in PBS for at least 20 min, incubated with primary antibody for at least 1 hr, washed three times with PBS and incubated with the respective secondary antibody for at least 45 min. Coverslips were washed again three times with PBS, incubated with 1 µg/ml Hoechst (Invitrogen) in PBS for 10 min and washed again. Coverslips were mounted on microscopic slides with Vectashield (Vector labs).

## Protein purification

Recombinant zz-His$_6$, zz-LAP1B(98-136)-His$_6$ wild-type and the respective 2E mutant (S108E, T124E) for EMSAs were produced in *E. coli* XL1(pBS161) (*Kutay et al., 2000*). Cells were lysed in 50 mM Tris pH 7.5, 700 mM NaCl, 2 mM MgCl$_2$, 35 mM Imidazole, 2 mM beta-mercaptoethanol, 1 mg/ml lysozyme, 10 µg/ml DNase I, the recombinant proteins retrieved by immobilized nickel affinity chromatography, and eluted with 50 mM Tris pH 7.5, 350 mM NaCl, 2 mM MgCl$_2$, 400 mM Imidazole, 2 mM beta-mercaptoethanol. The protein was rebuffered to 10 mM Tris pH 7.5, 200 mM NaCl, 1 mM EDTA, 1 mM DTT, 5% glycerol, 10% sucrose.

## Electrophoretic mobility shift assays (EMSAs) of DNA and mononucleosomes

DNA was produced as previously described (*Hanson et al., 2004*). In brief, a pUC19 plasmid carrying multiple copies of the Widom '601' sequence, each flanked by EcoRV restriction sites, was purified from an 8 l culture of transformed *E. coli* DH5α cells (NEB). The '601' fragments were released from the plasmid by digestion with EcoRV (NEB), isolated by incremental PEG precipitation, and further purified by ethanol-acetate precipitation and subsequent chloroform-phenol extraction. Histone octamers were prepared as described (*Luger et al., 1999*). We used wild-type histone proteins, except for H3.2 carrying a C110A mutation. Mononucleosomes were reconstituted through overnight logarithmic dialysis from high-salt (10 mM Tris pH 7.5, 1 mM EDTA 2 M KCl) into low-salt buffer (10 mM Tris pH 7.5, 1 mM EDTA, 10 mM KCl), yielding a 22.6 nM mononucleosome solution based on DNA absorbance.

EMSAs were performed with 10 nM mononucleosomes or 10 nM '601' DNA, preincubated with LAP1 protein derivatives at the indicated concentrations in a total volume of 10 µl for 10 min at 37°C (final buffer concentration 10 mM Tris pH 7.5, 1 mM EDTA, 150 mM KCl). Samples were put on ice, mixed with 5 µl ice-cold sample buffer (25% (w/v) sucrose, 0.01% (w/v) bromophenol blue) and loaded on a 5% poly-acrylamide (29:1, BioRad) 0.5 x TBE gel, migrated for 1 hr at 90 V in ice-cold 0.5 x TBE. DNA was stained with GelRed (Biotium) and imaged with a ChemiDoc MP(BioRad). Band intensities were quantified with ImageLab (BioRad). The intensities were normalized to the bands in the absence of added protein, and the averaged values of the replicates were plotted ± standard

deviation. The data were evaluated with a Hill equation $Intensity = 1 - \frac{1}{\left(1 + \left(\frac{K_d}{[protein]}\right)^n\right)}$ taking the standard deviation in account using IGOR (Wavemetrics).

### Image acquisition and live cell imaging

Confocal fluorescence images were acquired with LSM 880, LSM 780 microscope (ZEISS) or a Leica SP2 AOBS microscope. For all microscopes, 63 × 1.4 NA, oil plan-apochromat objectives were used. For time-lapse microscopy, cells were seeded into µ-slide 4-well ibidi chambers and incubated with 0.1 µM SiR-Hoechst 1 hr prior to the experiment. Images were acquired at a Visitron spinning disk Nikon Eclipse T1 microscope, using a 60 × 1.4 oil lens, in a 5% $CO_2$, 37°C chamber. 15 z-stacks were acquired every 5 min for 8 to 12 hr.

### Image processing and quantification

For quantification of GFP intensity at the nuclear rim, the integrated GFP density (IntDen) at the NE was determined with ImageJ. Nuclear contours were manually detected by placing individual ROIs (Region of Interests) around the NE. To quantify the NE aberration phenotype (*Figure 3A, B, C*), only cells binned for a window of similar fluorescence intensities were included into the analysis as indicated. IntDen GFP was normalized relative to the average binned GFP IntDen of cells expressing LAP1B-GFP only.

### FRAP

Cells were seeded into NuncTM Lab-TekTMII chambers, and transfected or induced one day before the experiment. For FRAP, either an LSM 710-FCS, LSM 780-FCS or LSM 880 Carl Zeiss microscope was used. Cells were kept in a humified chamber at 5% $CO_2$ at 37°C. Time-lapse movies were recorded with a 63 × 1.4 NA oil DIC plan-apochromat immersion lens, and images were taken with a 4 x zoom. Three pre-bleach values were acquired, and then a 3.3 × 0.66 (2.2 µm²) rectangle at the NE was bleached (with 100% 488 nm laser intensity, and a range of 80–200 scanning iterations to achieve 90% bleaching). The fluorescence recovery was measured every 4 s for 57 cycles. The acquired FRAP data was normalized and fitted by a double exponential using the easyFRAP Matlab tool (*Rapsomaniki et al., 2012*).

### Statistical methods

Graphs were generated using Prism, which was further used to calculate the unpaired t-test and p values less than 0.001 are indicated by four asterisks (****$p \leq 0.0001$), three asterisks (***$p \leq 0.001$); p values less than 0.01 by two asterisks (**$p \leq 0.01$); p values less than 0.05 by one asterisk (*$p \leq 0.05$); and p values higher than 0.05 are marked as 'not significant' (ns).

### Immunoblotting

Whole cell lysates were generated by resuspending cells in SDS sample buffer (75 mM Tris pH 6.8, 20% (v/v) glycerol, 4% (w/v) SDS, 50 mM DTT, 0.01% (w/v) bromophenole blue). Proteins were separated on 8–15% SDS gels and transferred to a nitrocellulose membrane (GE Healthcare) using the Trans-Blot SD Semi-Dry Transfer Cell (Bio-Rad). Membranes were incubated after blocking with 5% (w/v) dry milk in TBT with the primary antibody for at least 1 hr. As secondary antibodies, rabbit anti-mouse or goat anti-rabbit horseradish-peroxidase-conjugated antibodies (Sigma) were used. Signals were captured by exposure to film or detected using a Fusion (Vilber) or an Odyssey (LI-COR) imaging system.

## Acknowledgements

We thank Sumit Pawar, Annamaria Gamper, Jasmin van den Heuvel and Ivo Zemp for critical comments on the manuscript as well as the members of the Kutay lab for helpful discussions. Microscopy was performed on instruments of the ETHZ Microscopy Center Scope M. This work was funded by a grant of the Swiss National Science Foundation (SNSF) to U.K. (310030_184801), and by EPFL (BF).

## Additional information

### Funding

| Funder | Grant reference number | Author |
|---|---|---|
| Swiss National Science Foundation | 310030_184801 | Ulrike Kutay |
| EPFL | | Beat Fierz |

The funders had no role in study design, data collection and interpretation, or the decision to submit the work for publication.

### Author contributions

Naemi Luithle, Conceptualization, Data curation, Formal analysis, Investigation, Visualization, Methodology, Writing - original draft, Writing - review and editing; Jelmi uit de Bos, Formal analysis, Validation, Investigation, Visualization, Writing - review and editing; Ruud Hovius, Investigation, Visualization, Methodology, Writing - review and editing; Daria Maslennikova, Investigation, Visualization, Writing - review and editing; Renard TM Lewis, Resources, Investigation, Writing - review and editing; Rosemarie Ungricht, Conceptualization, Investigation, Visualization, Methodology; Beat Fierz, Conceptualization, Supervision, Writing - review and editing; Ulrike Kutay, Conceptualization, Data curation, Formal analysis, Supervision, Funding acquisition, Writing - original draft, Project administration, Writing - review and editing

### Author ORCIDs

Jelmi uit de Bos (iD) https://orcid.org/0000-0002-8807-3284
Ruud Hovius (iD) http://orcid.org/0000-0001-9258-6587
Ulrike Kutay (iD) https://orcid.org/0000-0002-8257-7465

### Decision letter and Author response

Decision letter https://doi.org/10.7554/eLife.63614.sa1
Author response https://doi.org/10.7554/eLife.63614.sa2

## Additional files

### Supplementary files

• Transparent reporting form

### Data availability

All data generated or analysed during this study are included in the manuscript and supporting files.

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

# Appendix 1

**Appendix 1—key resources table**

| Reagent type (species) or resource | Designation | Source or reference | Identifiers | Additional information |
|---|---|---|---|---|
| Cell line (*Homo sapiens*) | HeLa FlpIn T-REx LAP1B-EGFP | This paper | | See Materials and methods, *Cell lines, antibodies, and reagents* |
| Cell line (*Homo sapiens*) | HeLa FlpIn T-REx LAP2β-EGFP | This paper | | See Materials and methods, *Cell lines, antibodies, and reagents* |
| Cell line (*Homo sapiens*) | HeLa FlpIn T-REx LAP1C-EGFP | This paper | | See Materials and methods, *Cell lines, antibodies, and reagents* |
| Cell line (*Homo sapiens*) | HeLa FlpIn T-REx LAP1B(1-359)-EGFP | This paper | | See Materials and methods, *Cell lines, antibodies, and reagents* |
| Cell line (*Homo sapiens*) | HeLa FlpIn T-REx LAP1B(E482A)-EGFP | This paper | | See Materials and methods, *Cell lines, antibodies, and reagents* |
| Cell line (*Homo sapiens*) | HeLa FlpIn T-REx LAP1B(R563G)-EGFP | This paper | | See Materials and methods, *Cell lines, antibodies, and reagents* |
| Cell line (*Homo sapiens*) | HeLa FlpIn T-REx LAP1B(1-359, S108E, T124E)-EGFP | This paper | | See Materials and methods, *Cell lines, antibodies, and reagents* |
| Cell line (*Homo sapiens*) | HeLa FlpIn T-REx Tor1B-EGFP | This paper | | See Materials and methods, *Cell lines, antibodies, and reagents* |
| Cell line (*Homo sapiens*) | HeLa FlpIn T-REx Tor1B(E178Q)-EGFP | This paper | | See Materials and methods, *Cell lines, antibodies, and reagents* |
| Cell line (*Homo sapiens*) | HeLa FlpIn T-REx | (Hafner et al., 2014) DOI:10.1038/ncomms5397 | | Obtained from T. Maier (University Konstanz). |
| Cell line (*Homo sapiens*) | HeLa Kyoto | other | RRID:CVCL_1922 | Obtained from D. Gerlich (IMBA, Vienna). |
| Cell line (*Homo sapiens*) | HeLa Gromeier LMN A/C KO | This paper | | See Materials and methods, *Cell lines, antibodies, and reagents antibodies, and reagents* |
| Cell line (*Homo sapiens*) | HCT116 | other | RRID:CVCL_0291 | Obtained from B. Vogelstein (Johns Hopkins, Baltimore, USA) |
| Cell line (*Homo sapiens*) | HepG2 | other | RRID:CVCL_0027 | Obtained from S. Werner (Institute of Molecular Health Science, Zurich) |
| Transfected construct (human) | si-Control | Qiagen | Cat# 1027281 | Allstars siRNA |
| Transfected construct (human) | si-lamin A/C | Microsynth (**Hasan et al., 2006**) DOI: 10.1016/j.febslet.2006.01.039 | | 5'- CUGGACUU CCAGAAGAACA -3' |
| Transfected construct (human) | si-lamin B1 | Sigma (**Hasan et al., 2006**) DOI: 10.1016/j.febslet.2006.01.039 | | 5'- UUCCGCCUCA GCCACUGGAAAU -3' |

*Continued on next page*

*Appendix 1—key resources table continued*

| Reagent type (species) or resource | Designation | Source or reference | Identifiers | Additional information |
|---|---|---|---|---|
| Transfected construct (human) | si-lamin B2 | Microsynth | | 5'- ACAACUCGG ACAAGGAUC -3' |
| Transfected construct (human) | si-LAP1 | Qiagen/Microsynth | | 5'- CUCACUAAGU UUCCUGAGUUA- 3' |
| Transfected construct (human) | si-LULL1 | Qiagen/Microsynth | | 5'- CTGGTCCTGA CTGTTCTGCTA -3' |
| Transfected construct (human) | si-Tor1A | Qiagen/Microsynth | | 5'- CACCAAGTTAG ATTATTACTA-3' |
| Transfected construct (human) | si-Tor1B | Qiagen/Microsynth | | 5'- CTGTCGGAGT CTTCAATAATA-3' |
| Antibody | anti-β-actin (mouse monoclonal) | Sigma | Cat# A1978 RRID:AB_476692 | WB(1:40'000) |
| Antibody | anti-emerin (rabbit polyclonal) | Abcam | Cat# ab40688 RRID:AB_2100059 | WB(1:1000) |
| Antibody | anti-HA (mouse monoclonal) | Covance | Cat# MMS-101P RRID:AB_2314672 | IF(1:3000) WB(1:3000) |
| Antibody | anti-lamin A/C (mouse monoclonal) | ImmuQuest | Cat# IQ332 RRID:AB_10660272 | IF(1:200) WB (1:200) |
| Antibody | anti-lamin A/C (rabbit polyclonal) | Proteintech | Cat# 10298-1-AP RRID:AB_2296961 | IF(1:500) |
| Antibody | anti-lamin B1 (rabbit polyclonal) | Abcam | Cat# ab16048 RRID:AB_443298 | IF(1:2000) WB(1:1000) |
| Antibody | anti-lamin B2 (rabbit monoclonal) | Abcam | Cat# ab151735 RRID:AB_2827514 | IF (1:1000) WB(1:1000) |
| Antibody | anti-LAP1 (rabbit polyclonal) | Abcam | Cat# ab86307 RRID:AB_2206124 | IF(1:500) WB(1:300) |
| Antibody | anti-LBR (rabbit polyclonal) | Abnova | Cat# PAB15583 RRID:AB_10696691 | IF(1:1000) |
| Antibody | anti-mAB414 (mouse monoclonal) | Abcam | Cat# ab 24609 RRID:AB_448181 | IF(1:20000) |
| Antibody | anti-H3 (rabbit polyclonal) | Abcam | Cat# ab1791 RRID:AB_302613 | WB(1:5000) |
| Antibody | anti-Tor1A (rabbit polyclonal) | Abexxa | Cat# abx001683 | WB(1:500) |

*Continued on next page*

*Appendix 1—key resources table continued*

| Reagent type (species) or resource | Designation | Source or reference | Identifiers | Additional information |
|---|---|---|---|---|
| Antibody | anti-Tor1B (rabbit polyclonal) | antibodies-online | Cat# ABIN1860834 | WB(1:500) |
| Antibody | anti-α-tubulin (mouse monoclonal) | Sigma | Cat# T5168 RRID:AB_477579 | WB(1:20000) IF(1:20000) |
| Antibody | anti-GAPDH (mouse monoclonal) | Abcam | Cat# ab8245 RRID:AB_2107448 | WB(1:10000) |
| Antibody | anti-LULL1 (rabbit polyclonal) | This paper | | IF (1:500) See Materials and methods, *Cell lines, antibodies, and reagents* |
| Antibody | anti-SUN1 (rabbit polyclonal) | (*Sosa et al., 2012*) DOI: 10.1083/jcb.200904048 | | IF(1:1000) |
| Antibody | anti-SUN2 (rabbit polyclonal) | (*Turgay et al., 2010*) DOI: 10.1261/rna.2325911 | | IF(1:2000) |
| Antibody | anti-GFP (rabbit polyclonal) | (*Turgay et al., 2010*) DOI: 10.1261/rna.2325911 | | IF(1:1000) WB(1:1000) |
| Recombinant DNA reagent | pcDNA5/FRT/TO/LAP1B-EGFP | This paper | | See Materials and methods, *Molecular cloning* |
| Recombinant DNA reagent | pcDNA5/FRT/TO/LAP1C-EGFP | This paper | | See Materials and methods, *Molecular cloning* |
| Recombinant DNA reagent | pK7 LAP1B(1-72)-GST-EGFP | This paper | | LAP1: aa 1-72 See Materials and methods, *Molecular cloning* |
| Recombinant DNA reagent | pK7 LAP1B(1-183)-GST-EGFP | This paper | | LAP1: aa 1-183 See Materials and methods, *Molecular cloning* |
| Recombinant DNA reagent | pK7 LAP1B(98-136)-GST-EGFP | This paper | | LAP1: aa 98-136 See Materials and methods, *Molecular cloning* |
| Recombinant DNA reagent | pK7 LAP1B(184-337)-GST-EGFP | This paper | | LAP1: aa 184-337 See Materials and methods, *Molecular cloning* |
| Recombinant DNA reagent | pcDNA5/FRT/TO/LAP1B(Δ98-136)-EGFP | This paper | | LAP1: aa Δ98-136 See Materials and methods, *Molecular cloning* |
| Recombinant DNA reagent | pcDNA5/FRT/TO/LAP1B(1-359)-2E-EGFP | This paper | | LAP1: aa 1-359, S108E, T124E See Materials and methods, *Molecular cloning* |
| Recombinant DNA reagent | pcDNA5/FRT/TO/LAP1B(1-359)-EGFP | This paper | | LAP1: aa 1-359, See Materials and methods, *Molecular cloning* |
| Recombinant DNA reagent | pcDNA5/FRT/TO/ Tor1A-cTAP | This paper | | See Materials and methods, *Molecular cloning* |
| Recombinant DNA reagent | pcDNA5/FRT/TO/ Tor1A-cTAP | This paper | | See Materials and methods, *Molecular cloning* |

*Continued on next page*

*Appendix 1—key resources table continued*

| Reagent type (species) or resource | Designation | Source or reference | Identifiers | Additional information |
|---|---|---|---|---|
| Recombinant DNA reagent | pcDNA5/FRT/ TO/ Tor1B-cTAP | This paper | | See Materials and methods, *Molecular cloning* |
| Recombinant DNA reagent | pcDNA5/FRT/TO/ Tor1B(E178Q)-cTAP | This paper | | See Materials and methods, *Molecular cloning* |
| Recombinant DNA reagent | pcDNA5/FRT/TO/ LAP1B(R563G)-EGFP | This paper | | See Materials and methods, *Molecular cloning* |
| Recombinant DNA reagent | pcDNA5/FRT/TO/ LAP1B-E482A-EGFP | This paper | | See Materials and methods, *Molecular cloning* |
| Recombinant DNA reagent | pcDNA5/FRT/TO/ Tor1B-mRFP | This paper | | See Materials and methods, *Molecular cloning* |
| Recombinant DNA reagent | pcDNA5/FRT/TO/ Tor1B(E178Q)-mRFP | This paper | | See Materials and methods, *Molecular cloning* |
| Recombinant DNA reagent | pcDNA5/FRT/TO/ Tor1B-EGFP | This paper | | See Materials and methods, *Molecular cloning* |
| Recombinant DNA reagent | pcDNA5/FRT/TO/ Tor1B(E178Q)-EGFP | This paper | | See Materials and methods, *Molecular cloning* |
| Recombinant DNA reagent | pcDNA5/FRT/TO/ LAP1B(S108E, T124E)-EGFP | This paper | | S108E, T124E See Materials and methods, *Molecular cloning* |
| Recombinant DNA reagent | pcDNA5/FRT/TO/ LAP1B(S108E, T124E, R563G)-EGFP | This paper | | S108E, T124E, R563G See Materials and methods, *Molecular cloning* |
| Recombinant DNA reagent | pK7-GST-GFP | (*Ungricht et al., 2015a*) DOI: 10.1007/978-1-4939-3530-7_28 | | See Materials and methods, *Molecular cloning* |
| Recombinant DNA reagent | pK7 NLS-GST-EGFP | (*Erkmann et al., 2005*) DOI: 10.1091/mbc.e04-11-1023 | | See Materials and methods, *Molecular cloning* |
| Recombinant DNA reagent | pK7 LAP1(1-121)-GST-EGFP | This paper | | LAP1: aa 1-121, See Materials and methods, *Molecular cloning* |
| Recombinant DNA reagent | pK7 LAP1B(73-183)-GST-EGFP | This paper | | LAP1: aa 73-183, See Materials and methods, *Molecular cloning* |
| Recombinant DNA reagent | pK7 LAP1(122-183)-GST-EGFP-NLS | This paper | | LAP1: aa 122-183, See Materials and methods, *Molecular cloning* |
| Recombinant DNA reagent | pK7 LAP1B(73-337)-GST-EGFP | This paper | | LAP1: aa 73-337, See Materials and methods, *Molecular cloning* |
| Recombinant DNA reagent | pK7 LAP1(122-337)-GST-EGFP | This paper | | LAP1: aa 122-337, See Materials and methods, *Molecular cloning* |

*Continued on next page*

*Appendix 1—key resources table continued*

| Reagent type (species) or resource | Designation | Source or reference | Identifiers | Additional information |
|---|---|---|---|---|
| Recombinant DNA reagent | pK7 LAP1B(1-337)-GST-EGFP | This paper | | LAP1: aa 1-337, See Materials and methods, *Molecular cloning* |
| Recombinant DNA reagent | pK7 LAP1(1-72) GST-EGFP-NLS | This paper | | LAP1: aa 1-72, See Materials and methods, *Molecular cloning* |
| Recombinant DNA reagent | pK7 LAP1(122-183)-GST-EGFP-NLS | This paper | | LAP1: aa 122-183, See Materials and methods, *Molecular cloning* |
| Recombinant DNA reagent | pK7 LAP1(73-337) GST-EGFP-NLS | This paper | | LAP1: aa 73-337, See Materials and methods, *Molecular cloning* |
| Recombinant DNA reagent | pK7 LAP1(184-337) GST-EGFP-NLS | This paper | | LAP1: aa 184-337, See Materials and methods, *Molecular cloning* |
| Recombinant DNA reagent | pEGFPN3 LAP1B -EGFP | This paper | | See Materials and methods, *Molecular cloning* |
| Recombinant DNA reagent | pEGFPN3 EMD-EGFP | This paper | | See Materials and methods, *Molecular cloning* |
| Recombinant DNA reagent | pEGFPN3 LAP2β-EGFP | (*Ungricht et al., 2015a*) DOI: 10.1007/978-1-4939-3530-7_28 | | See Materials and Methods, *Molecular cloning* |
| Recombinant DNA reagent | pEGFPN3 LEM2-EGFP | This paper | | See Materials and Methods, *Molecular cloning* |
| Recombinant DNA reagent | pEGFPN3 SUN1-EGFP | (*Turgay et al., 2010*) DOI: 10.1261/rna.2325911 | | See Materials and Methods, *Molecular cloning* |
| Recombinant DNA reagent | pEGFPN3 SUN2-EGFP | (*Turgay et al., 2010*) DOI: 10.1261/rna.2325911 | | See Materials and Methods, *Molecular cloning* |
| Recombinant DNA reagent | pEGFPN3-SPAG4-GFP | (*Turgay et al., 2010*) DOI: 10.1261/rna.2325911 | | See Materials and Methods, *Molecular cloning* |
| Recombinant DNA reagent | LAP1(98-136)-SPAG4-EGFP | This paper | | See Materials and Methods, *Molecular cloning* |
| Recombinant DNA reagent | LAP1(98-136_2E)-SPAG4-EGFP | This paper | | See Materials and Methods, *Molecular cloning* |
| Recombinant DNA reagent | LAP1(98-136_3E)-SPAG4-EGFP | This paper | | See Materials and Methods, *Molecular cloning* |
| Recombinant DNA reagent | LULL1-HASt | This paper | | See Materials and Methods, *Molecular cloning* |
| Recombinant DNA reagent | pCMV/ER/myc EGFP-KDEL | (*Ungricht et al., 2015a*) DOI: 10.1007/978-1-4939-3530-7_28 | | See Materials and Methods, *Molecular cloning* |
| Recombinant DNA reagent | pEGFPN3 LAP1B(72-end)-EGFP | This paper | | LAP1: aa 72-583, See Materials and Methods, *Molecular cloning* |
| Recombinant DNA reagent | pEGFPN3 LAP1B(184-end)-EGFP | This paper | | LAP1: aa 184-583, See Materials and Methods, *Molecular cloning* |
| Recombinant DNA reagent | pEGFPN3 LAP1C(Δ122-136) EGFP | This paper | | LAP1: aa Δ122-136, See Materials and Methods, *Molecular cloning* |

*Appendix 1—key resources table continued*

| Reagent type (species) or resource | Designation | Source or reference | Identifiers | Additional information |
|---|---|---|---|---|
| Recombinant DNA reagent | pcDNA5/FRT/TO/LAP1C(T124E)-EGFP | This paper | | See Materials and Methods, *Molecular cloning* |
| Recombinant DNA reagent | pcDNA5/FRT/TO/LAP1B(184-end R563G)-EGFP | This paper | | LAP1: aa 184-583, See Materials and Methods, *Molecular cloning* |
| Recombinant DNA reagent | pEGFPN3 LAP1(R563G)-EGFP | This paper | | See Materials and Methods, *Molecular cloning* |
| Recombinant DNA reagent | pK7 LAP1(1-97)-GST-GFP | This paper | | LAP1: aa 1-97, See Materials and Methods, *Molecular cloning* |
| Recombinant DNA reagent | pK7 LAP1(137-337)-GST-GFP | This paper | | LAP1: aa 137-337, See Materials and Methods, *Molecular cloning* |
| Recombinant DNA reagent | pQE60zz-His$_6$ | (*Sosa et al., 2012*) DOI: 10.1016/j.cell.2012.03.046 | | See Materials and Methods, *Molecular cloning* |
| Recombinant DNA reagent | pQE60zz-LAP1B(98-136)-His$_6$ | This paper | | LAP1: aa 98-136, See Materials and Methods, *Molecular cloning* |
| Recombinant DNA reagent | pQE60zz-His$_6$ zz-LAP1B(98-136)-2E His$_6$ | This paper | | LAP1: aa 98-136, S108E, T124E See Materials and Methods, *Molecular cloning* |
| Recombinant DNA reagent | pC2P | (*Welte et al., 2019*) DOI: 10.1101/gad.328492.119 | | |
| Recombinant DNA reagent | pC2P-gLMNA/C | This paper | | Protospacer: 5'- CACCG GGTGGCGCGCC GCTGGGACG -3' See Materials and Methods, *Generation of LMN A/C knockout cells* |
| Chemical compound, drug | SIR-Hoechst | Spirochrome | Cat# SC007 | |
| Chemical compound, drug | Hoechst | Invitrogen | Cat# 63493 | |
| Chemical compound, drug | DTT | Applichem | Cat# A1101 | |
| Chemical compound, drug | nocodazole | Sigma-Aldrich | Cat# M1404 | |
| Chemical compound, drug | thymidine | Sigma-Aldrich | Cat #T1895 | |
| Chemical compound, drug | tetracycline | Invitrogen | Cat# 550205 | |

