## [Decision Letter]

**Acceptance summary:**

The integral inner nuclear membrane protein LAP1 is an established activator of the lumenal AAA ATPase Torsin. In this manuscript the authors uncover an unexpected mechanism in which Torsin influences the chromatin binding properties of LAP1 in a manner requiring its ATPase activity. These new findings support a key role for the LAP1-Torsin axis in regulating mitotic nuclear envelope remodeling.

**Decision letter after peer review:**

Thank you for submitting your article "Torsin ATPases influence chromatin interaction of the Torsin regulator LAP1" for consideration by *eLife*. Your article has been reviewed by three peer reviewers, including Megan C King as the Reviewing Editor and Reviewer #1, and the evaluation has been overseen by Anna Akhmanova as the Senior Editor.

The reviewers have discussed the reviews with one another and the Reviewing Editor has drafted this decision to help you prepare a revised submission.

Summary:

This manuscript presents an interesting novel idea: that the inner nuclear membrane (INM) protein LAP1, known as a regulator of Torsins, is itself regulated by Torsins. Luithle and colleagues describe a novel chromatin binding domain in the in the nucleoplasmic region of LAP1 that can directly bind DNA and functions at least in part to regulate LAP1's retention at the INM. The authors also identify two lamin interaction domains and show that these interactions further reinforce the anchorage of LAP1 at the INM. The authors tie the chromatin binding by LAP1 to its ability, when over-expressed, to induce altered nuclear morphology, defects in membrane removal during mitosis and features consistent with chromosome segregation defects. They discover that Torsin over-expression can modulate the phenotypes tied to LAP1 over-expression and therefore mimic the deletions/mutations that abrogate chromatin binding; this requires a functional Torsin-LAP1 interaction and TorsinB ATPase activity. Thus, for the first time, the authors describe a direct role for Torsins in promoting mitotic integrity via LAP1, suggesting that catalytically active Torsin, which is limiting, is required to release LAP1 from chromatin during mitosis.

All three reviewers were enthusiastic about the study, highlighting that it was well-written, presented data of consistently high quality, and would be of interest to a broad readership. The reviewers agreed that the manuscript reveals important new information on the mutual regulation of Torsins and LAP1 and provides a conceptual advance for our understanding of mitotic nuclear envelope breakdown via an unexpected mechanism operating across the inner nuclear membrane. The authors had some suggestions for a revised manuscript, most of which can be addressed with edits to the text.

Essential revisions:

1) The reviewers were somewhat troubled by the interpretation that "…no post mitotic NE aberrations were observed in Tor1B(E178Q) expressing cells…", as shown in Figure 6B, C. This is confusing and seemingly contradicts the working model saying that increased LAP1 levels relative to active Tor1 lead to NE aberrations, in turn generating binucleated cells. As there is a certain degree of functional redundancy between Torsins (Laudermilch et al., 2016 and Tanabe et al., 2016), could this be due to the presence of endogenous TorsinA/B? Arguably, the most definitive experiment to support their central conclusion would be to show that a TorA/B knockout, or their co-depletion, results in the same phenotype as LAP1 overexpression considering that only TorA/B (but none of the other Torsins) act on Lap1 (Zhao et al., 2013). Or is even a TorB KO alone sufficient to cause similar effects?

2) The analyses strongly rely on the quantification of nuclear defects, but in some figures differences in nuclear defect severity are not so clear in the immunofluorescence images. How was severity of nuclear morphology defects quantified? From the images alone correlation with expression level is not so clear. The way that nuclear aberrations were quantified should be explained carefully in the text and/or legends.

3) Please address if and how the expression level of INM proteins tested was controlled, as this seems to be relevant for the phenotype (see Figure 1D)? Particularly SUN1 seems to be weakly expressed.

4) How did authors make sure that the cells had no nuclear aberrations before time lapse imaging was started? Did these cells never go through mitosis in the presence of the GFP-LAP1 before? How long do the cells survive in the presence of overexpressed LAP1? Authors may consider to show some of the data presented in Figure 5 (e.g. bi-nucleated cells) already here, to address cell fate upon LAP1 overexpression.

5) One important aspect of the proposed model, not addressed in the study and not even discussed in the manuscript, is the fact that Torsin's activity on LAP1 has to be regulated during the cell cycle. LAP1 is tethered to the nuclear periphery to a great part by its interaction with chromatin in interphase cells, which contain active Torsin. In mitosis Torsin is required to release LAP1 from chromatin. Is Torsin regulated or do chromatin changes in mitosis contribute to the Torsin-mediated LAP1 release? The authors should at least address this point in the Discussion. Along these lines, although the authors have not been able to dissect the phosphorylation of the CBR in detail to this point, it is worth asking how the phosphodead/phosphomimetic mutations influence the chromatin association of the CBR alone. Similarly, one wonders if the "generic" phosphorylation status of LAP1B (as visualized on a PhosTag gel/blot, for example) is altered during mitosis and whether this is influenced by dominant-negative forms of Torsin.

6) Although not absolutely essential, evidence for their speculation that the (hetero)oligomeric state of the Torsin/cofactor system influences their findings would be important for the Torsin field. The authors could test one key aspect comparatively easily: would a back interface Torsin mutant (Chase et al., 2017) selectively perturbing the Torsin homo-oligomer, but not the LAP1 interaction, fail or succeed in reverting the Lap1 overexpression phenotype? The result could materially inform our mechanistic understanding of this novel transmembrane relay mechanism.

7) The relatively low level of LAP1B expression relative to LAP1C expression (and weaker association of LAP1C with chromatin) raises the possibility that LAP1C also competes with LAP1B for other binding partners, including Torsins. Although the authors note that LAP1C can also induce nuclear aberrations (albeit much more weakly than LAP1B) one wonders if a functional LAP1C-Torsin complex might serve a function that is disrupted by LAP1B over-expression. Is there any evidence for such crosstalk?

---

## [Author Response]

Essential revisions:1) The reviewers were somewhat troubled by the interpretation that "…no post mitotic NE aberrations were observed in Tor1B(E178Q) expressing cells…", as shown in Figure 6B, C. This is confusing and seemingly contradicts the working model saying that increased LAP1 levels relative to active Tor1 lead to NE aberrations, in turn generating binucleated cells. As there is a certain degree of functional redundancy between Torsins (Laudermilch et al., 2016 and Tanabe et al., 2016), could this be due to the presence of endogenous TorsinA/B? Arguably, the most definitive experiment to support their central conclusion would be to show that a TorA/B knockout, or their co-depletion, results in the same phenotype as LAP1 overexpression considering that only TorA/B (but none of the other Torsins) act on Lap1 (Zhao et al., 2013). Or is even a TorB KO alone sufficient to cause similar effects?

We apologize that we have missed to provide a rational explanation for the above statement. We previously reported that post-mitotic NE aberrations are induced when membranes are artificially tethered to chromatin throughout mitosis (Champion et al., 2019). Notably, NE aberrations also arise when membranes are only tethered to chromatin from metaphase onwards, indicating that the precocious presence of membranes on chromatin during mitotic exit is the decisive parameter for this phenotype. In contrast, chromosome segregation errors and binucleation are caused by membrane chromatin contacts during pro-(meta)phase, and not observed if membranes are only tethered during metaphase, i.e. after chromosome alignment at the spindle equator.

While we always observed both NE aberrations and chromosome segregation errors/ binucleation when we induced LAP1B expression, Tor1B(E178Q) expression robustly induced chromosome segregation errors but failed to cause the prominent NE aberrations that we observed upon LAP1 overexpression, as highlighted by our note in the text. We don’t think that this contradicts our working model for two reasons. Firstly, when we overexpress LAP1B-GFP, GFP-positive membranes cover almost the entire mitotic chromatin mass in metaphase, while we observe fewer membrane patches at metaphase chromatin in presence of Tor1B(E178Q) at endogenous levels of LAP1. The latter seems sufficient to cause chromosome segregation defects but not NE aberrations. Secondly, it is possible that the membrane patches induced by Tor1B(E178Q) at endogenous levels of LAP1 are not as stably bound to chromatin due to their lower avidity, are more dynamically remodelled, or even eventually resolved, perhaps helped by the endogenous Torsins, as suggested by the reviewers. Notably, for chromosome segregation errors prometaphase is the decisive time window, while it is later mitosis (anaphase) for NE aberrations, providing a rationale as to why chromosome segregation errors are more likely to be consistent between both treatments.

We have now revised this section to read: “Notably, no severe post-mitotic NE aberrations were observed in Tor1B(E178Q) expressing cells, which might be explained by fewer, weaker and/or less persistent chromatin interactions of endogenous LAP1 in anaphase compared to overexpressed LAP1. We know from our previous work that late mitosis is indeed critical for the manifestation of this phenotype (Champion et al., 2019)”.

As suggested by the reviewers, we had tested whether depletion or knockout of Torsins induces a similar phenotype as Tor1B(E178Q) expression, however these data remained inconclusive. Firstly, we had generated TOR1A knock-out cells by CRSIPR/Cas9 and observed a high level of bi- and multinucleation in different clones generated with different guides as well as some nuclear shape abnormalities, consistent with our other observations. However, as binucleation is a phenotype that cannot be rescued, we performed a number of controls that then indicated that our CRISPR approach itself increased cell division defects when designed for unrelated targets, as long as the chosen guides found a genomic target. Thus, this data remained inconclusive. Similarly, we performed RNAi experiments to deplete Tor1A and Tor1B, the most abundant Torsins in HeLa cells. While we observed an increase in binucleation in the chosen HeLa cell line from 2% to about 6%, we feel that this data is too weak on its own, and we prefer to further investigate this issue in the future. However, we wish to emphasize that it is conceivable that the formation of Torsin-LAP1 oligomers might contribute to enhanced chromatin binding of LAP1, as already mentioned in the Discussion. In such a scenario, depletion of Torsins must not necessarily mimic the effect of Tor1B(E178Q) or LAP1B expression. Thus, this point remains to be addressed by other methods in the future.

2) The analyses strongly rely on the quantification of nuclear defects, but in some figures differences in nuclear defect severity are not so clear in the immunofluorescence images. How was severity of nuclear morphology defects quantified? From the images alone correlation with expression level is not so clear. The way that nuclear aberrations were quantified should be explained carefully in the text and/or legends.

In all figures, the intensity of the GFP signal allows for a direct comparison of expression levels since imaging of cells from the same experiment was performed at identical settings. At all indicated instances, we have additionally binned cells of similar expression levels for quantitative analyses. Nuclear shape aberrations were quantified in Figure 3, and the classification of assigned categories is shown in Figure 3B. As we did not use machine learning or another image analysis tool for quantification of NE aberrations, we have digitalized the 5 phenotypic classes into only two final bins, i.e. “yes” and “no” categories. We apologize that we had not described these aspects sufficiently. We have now added a section on quantification of GFP intensities and nuclear shape aberrations to Materials and methods (“Image processing and quantification”), and extended the corresponding figure legends.

3) Please address if and how the expression level of INM proteins tested was controlled, as this seems to be relevant for the phenotype (see Figure 1D)? Particularly SUN1 seems to be weakly expressed.

Please, see also above. In brief, the expression levels were quantified by integrating the NE GFP signal of individual nuclei in the fluorescent images. We think that there might have been some confusion on the quantification provided in former Figure 1E (the panel formerly at the right side of the immunoblots), which actually showed the quantification of the fluorescent images in Figure 1D. We have now rearranged the panels to hopefully address the point of the reviewers and revised the corresponding legend.

We agree that SUN1 was only weakly expressed in the cells presented in Figure 1A. We have repeated the experiment, now also taking images at 24 h, and replaced the panel. This demonstrates that even at higher expression levels SUN1 does not induce phenotypes similar to LAP1B. It also highlights that LAP1B-induced NE aberrations are only apparent in the vast majority of cells at 48 h after transient transfection (see also next point).

4) How did authors make sure that the cells had no nuclear aberrations before time lapse imaging was started? Did these cells never go through mitosis in the presence of the GFP-LAP1 before? How long do the cells survive in the presence of overexpressed LAP1? Authors may consider to show some of the data presented in Figure 5 (e.g. bi-nucleated cells) already here, to address cell fate upon LAP1 overexpression.

For these time-lapse experiments, we used tetracycline-inducible cell lines for which we can control the timing of LAP1B induction. As we had observed that the NE was smooth in most LAP1B-expressing cells 24 h after induction but aberrant after 48 h (see also new Figure 1A), we started the time-lapse imaging after 24 h. To address the point of the reviewers, we now better describe the experimental timeline in the corresponding figure legend. Notably, the time-lapse microscopy experiment shown in Figure 1B shows that the NE is smooth in LAP1B-expressing cells before mitosis but ruffled after mitosis, in contrast to LAP2b-GFP-expressing cells, demonstrating that the NE aberrations manifest during mitotic exit. We now also include corresponding video files.

We have not performed any cell survival assay but observed severe multinucleation and flower ball shaped giant nuclei after 72 h of LAP1B overexpression.

Without a marker for cellular contours, we cannot indicate binucleation in Figure 1. Taking up the suggestion of the reviewers, we now refer to a pair of closely juxtaposed nuclei in the legend to Figure 1D as a sign of potential binucleation and refer the reader to later parts of the manuscript.

5) One important aspect of the proposed model, not addressed in the study and not even discussed in the manuscript, is the fact that Torsin's activity on LAP1 has to be regulated during the cell cycle. LAP1 is tethered to the nuclear periphery to a great part by its interaction with chromatin in interphase cells, which contain active Torsin. In mitosis Torsin is required to release LAP1 from chromatin. Is Torsin regulated or do chromatin changes in mitosis contribute to the Torsin-mediated LAP1 release? The authors should at least address this point in the Discussion. Along these lines, although the authors have not been able to dissect the phosphorylation of the CBR in detail to this point, it is worth asking how the phosphodead/phosphomimetic mutations influence the chromatin association of the CBR alone. Similarly, one wonders if the "generic" phosphorylation status of LAP1B (as visualized on a PhosTag gel/blot, for example) is altered during mitosis and whether this is influenced by dominant-negative forms of Torsin.

It is currently unclear whether, and if so how, the activity of Torsins towards LAP1 is regulated during the cell cycle. As we already stated in the Discussion, “we do not necessarily assume that Torsins have evolved to liberate LAP1 from chromatin for open mitosis or that they possess a function dedicated to mitosis. […] We rather consider it likely that mitosis is perturbed as a consequence of a perturbed LAP1-Torsin interplay during interphase that manifests in the subsequent mitosis.” In light of this, we actually do not think that one must necessarily assume any cell cycle-specific regulation.

We really appreciate the reasoning of the reviewers on the chromatin changes and have introduced a sentence in our Discussion, to read: “Finally, mitotic changes in chromatin may also promote LAP1 dissociation during mitosis, and these changes might be affected by perturbation of the LAP1-Torsin axis.”

Regarding the effect of the phospho-mimetic mutants of the CBR, these were tested in altogether three settings: Firstly, in Figure 2E, for their contribution to NE targeting and mitotic chromatin association of the isolated CBR in the context of the CBR-SPAG4 fusion protein, in Figure 2F, for their effect on the ability of LAP1B(1-359) to induce NE aberrations and mitotic chromatin association of membranes, and thirdly, in FRAP experiments measuring LAP1 mobility at the INM (Figure 2—figure supplement 3 and Figure 5). We have not performed an in-depth analysis of a corresponding phospho-deficient mutant. And while we have observed mitotic phosphorylation of LAP1B in mitosis (visualized by changes in migration in normal SDS page and of some sites by mass spectrometry), we have not studied the influence of Tor1B(E178Q) expression. These are excellent suggestions for our future studies.

6) Although not absolutely essential, evidence for their speculation that the (hetero)oligomeric state of the Torsin/cofactor system influences their findings would be important for the Torsin field. The authors could test one key aspect comparatively easily: would a back interface Torsin mutant (Chase et al., 2017) selectively perturbing the Torsin homo-oligomer, but not the LAP1 interaction, fail or succeed in reverting the Lap1 overexpression phenotype? The result could materially inform our mechanistic understanding of this novel transmembrane relay mechanism.

We fully agree that this would be very informative, and we have indeed tried to exploit this mutant in the experimental setup used in Figure 3A. The corresponding Tor1B(G258D) mutant did not fully rescue LAP1B-induced NE aberrations in contrast to the wild-type enzyme (our unpublished observations). However, in comparison to the Tor1B(E178Q) mutant, the NE aberrations seemed less severe upon expression of Tor1B(G258D). Thus, this mutant behaves like a “hypomorph” in our assay and we thus find it hard to infer a mechanistic conclusion from the observed “intermediate” phenotype.

7) The relatively low level of LAP1B expression relative to LAP1C expression (and weaker association of LAP1C with chromatin) raises the possibility that LAP1C also competes with LAP1B for other binding partners, including Torsins. Although the authors note that LAP1C can also induce nuclear aberrations (albeit much more weakly than LAP1B) one wonders if a functional LAP1C-Torsin complex might serve a function that is disrupted by LAP1B over-expression. Is there any evidence for such crosstalk?

It is very likely that LAP1B and LAP1C share additional binding partners, one example being nuclear lamins. However, as LAP1B presents an N-terminal extension of LAP1C, we would expect that most of the functions of LAP1C should also be covered by the long isoform. Thus, rather than LAP1C having a function that is disturbed by LAP1B overexpression, we consider it more likely that LAP1B interacts with additional factors and may have additional functions. So far, we have not performed any differential proteomics of LAP1B and LAP1C.